# BoostAdapter: Improving Vision-Language Test-Time Adaptation via Regional Bootstrapping

**Taolin Zhang**[1]    **Jinpeng Wang** [1]    **Hang Guo** [1]
**Tao Dai**[*2]    **Bin Chen** [3]    **Shu-tao Xia** [1,4]

[1] Tsinghua University    [2] Shenzhen University
[3] Harbin Institute of Technology    [4] PengCheng Laboratory
https://github.com/taolinzhang/BoostAdapter

## Abstract

Adaptation of pretrained vision-language models such as CLIP to various downstream tasks have raised great interest in recent researches. Previous works have proposed a variety of test-time adaptation (TTA) methods to achieve strong generalization without any knowledge of the target domain. However, existing training-required TTA approaches like TPT necessitate entropy minimization that involves large computational overhead, while training-free methods like TDA overlook the potential for information mining from the test samples themselves. In this paper, we break down the design of existing popular training-required and training-free TTA methods and bridge the gap between them within our framework. Specifically, we maintain a light-weight key-value memory for feature retrieval from instance-agnostic historical samples and instance-aware boosting samples. The historical samples are filtered from the testing data stream and serve to extract useful information from the target distribution, while the boosting samples are drawn from regional bootstrapping and capture the knowledge of the test sample itself. We theoretically justify the rationality behind our method and empirically verify its effectiveness on both the out-of-distribution and the cross-domain datasets, showcasing its applicability in real-world situations.

## 1    Introduction

Vision Language models [49, 16, 23–25, 7] have shown incredible performance in downstream vision tasks [1], such as classification [29, 55, 54, 8], generation [20, 38, 9] and recognition [46, 47]. Among these models, CLIP [36] has been trained with large-scale noisy image-text pairs and can generalize well in zero-shot recognition tasks. The key idea behind CLIP is modality alignment during training and similarity comparison during testing for classification. However, CLIP suffers from domain shift problems during test-time inference. In the presence of out-of-distribution issues [27, 43, 12] that commonly appear in real-world scenarios, CLIP may fail to effectively align the feature across modalities, leading to performance degradation.

Test-time adaptation (TTA) has been widely explored in recent approaches [43, 15, 41, 17] to mitigate misalignment issues and improve performance in downstream tasks. Current mainstream TTA methods can be divided into training-required methods and training-free methods, as depicted in Figure. 1a and Figure. 1b. Training-required approaches [43, 41, 39] adjust model parameters or learnable prompts based on self-supervised objectives like entropy and increase the prediction confidence of model for distribution adaptation. TPT [41] applies entropy minimization to the vision-language model first. Furthermore, inspired by consistency regularization, TPT performs information mining

---

*Correspongding author: Tao Dai (daitao.edu@gmail.com)

38th Conference on Neural Information Processing Systems (NeurIPS 2024).

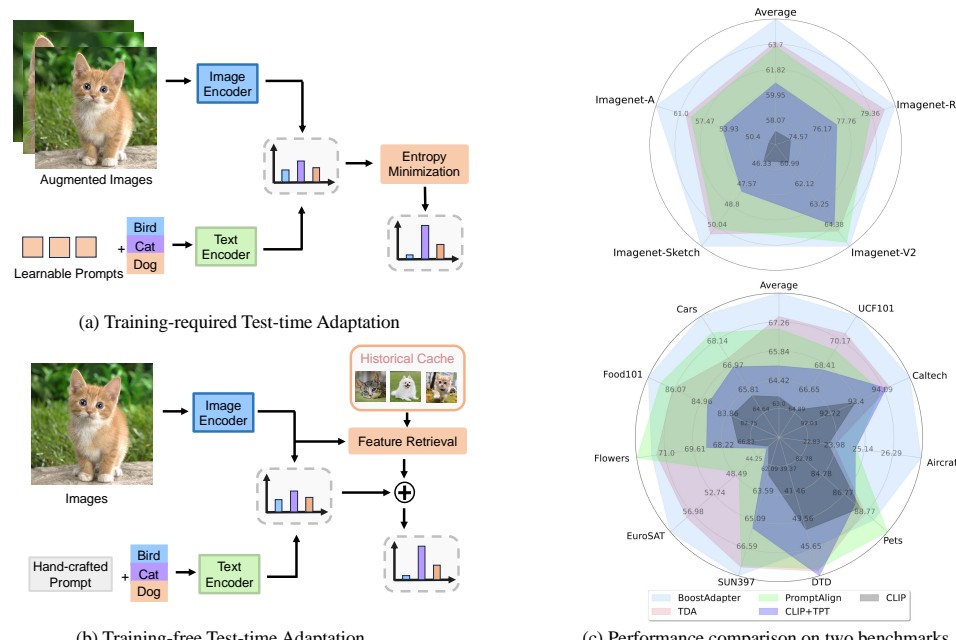

(a) Training-required Test-time Adaptation

(b) Training-free Test-time Adaptation

(c) Performance comparison on two benchmarks

Figure 1: (a) Existing training-required TTA methods utilize self-supervised objective like entropy minimization for better generalization. (b) Existing training-free TTA methods perform feature retrieval on the historical samples to adjust the model prediction. (c) Performance comparison on the Out-of-Distribution benchmark and Cross-Datasets benchmark.

from the test sample itself by random regional cropping in a self-bootstrapping style. However, training-required methods require gradient descent that is time-consuming with large training over-head, which prevents them from being applied in computationally limited situations. Training-free approaches [15, 52, 17] utilize memory networks, cache, or prototypes to store information regarding target samples and distributions, which is then used to adaptively modify the model's prediction. For example, TDA [17] leverages historical samples from the test data steam to build a dynamic key-value cache. It updates the prior knowledge encoded in CLIP through feature retrieval and output prediction based on the similarity between the test sample and the high-quality data stored in the memory bank. However, existing training-free approaches only consider interaction with other historical samples in the cache and do not effectively exploit the information within the test sample itself. This limitation prevents them from performing well especially in tasks that require fine-grained information.

Both of these approaches demonstrate excellent performance in enhancing the robustness of vision-language models to unknown distributions. However, the connection between them remains unclear. In this paper, we aim to answer three questions: (1) How are training-required methods like TPT and training-free methods like TDA connected? (2) How can we combine these two methods based on their shared nature? (3) Does vision-language models benefit from the combination of these methods?

In order to answer these questions, we first consider that the augmented images of test samples form a regional bootstrapping distribution of the original data. By filtering out the noisy augmentations based on mutual information with the predefined CLIP text embedding clusters, we can obtain a **boosting distribution** from which high-quality samples close to the target clusters can be drawn. Based on this, we delve into the connection between the target operations over the boosting distribution, *i.e.*, cross-entropy optimizations and cache classifier, which reveals the shared nature between entropy-based and cache-based methods. Specifically, we pinpoint that with the samples derived from the bootstrapping distribution, entropy minimization over them performs equivalently to feature retrieval from the cache consisting of them. Motivated by this analysis, we propose a brand-new adaptation strategy, dubbed **BoostAdapter**, to improve training-free adapters by incorporating the samples derived from the boosting distribution to the memory bank. Particularly, the cache in BoostAdapter consists of instance-agnostic historical samples filtered from the test data stream, along with instance-aware boosting samples generated through regional bootstrapping from the sample itself. The interactions between intra-sample and cross-sample operations make BoostAdapter effective and efficient by

incorporating the idea of information mining from training-required methods while maintaining the efficiency of training-free methods. Theoretical analyses and empirical results are also provided to validate the effectiveness of BoostAdapter.

To summarize, we make the following contributions in this paper.

- We first discuss the relationship between training-required and training-free methods in test-time adaptation and establish connections between them.
- We propose BoostAdapter, a brand new adaptation strategy in test-time adaptation of vision-language models, which improves training-free adapters by introducing high-quality samples from regional bootstrapping into the memory.
- We theoretically derive target domain error bound of BoostAdapter and shows that BoostAdapter benefit from incorporating self-bootstrapping data.
- Extensive experiments conducted over two benchmark demonstrate the superior performance of BoostAdapter under test-time adaptation settings.

## 2  Related Works

**Vision-Language Models**  have shown remarkable potential in generalization by contrastive pre-training over amounts of text-image pairs [16, 36, 24, 25] . One typical work is CLIP [36], which benefits from the alignment of 400 million curated image-text pairs and predicts the most relevant text description for a given image based on cosine similarity. Adapting CLIP to the downstream applications has attracted much attention and has been widely explored in recent approaches [55, 54, 52, 26, 56, 30]. CoOp [55] introduces learnable prompts [22, 51, 50, 28] and CoCoOp [54] conditions the text prompts on image embedding for better generalization. Maple [18] performs prompting for both vision and language branches and improves the alignment of the embedding between modalities. These approaches have demonstrated significant performance enhancements, but they still require few training data from the target domain. In contrast, we focus on test-time adaptation where there is no information about the target distribution and aim to generalize the model to any unknown scenarios.

**Training-required Test-time Adaptation**  updates partial wights of the model like prompts [41, 39] or BN layer [43] with self-supervised objectives that benefit the downstream tasks without requiring additional training data. Tent [43] reduces generalization error on shifted data by test-time entropy minimization. For vision-language models, Test-time prompt tuning (TPT) [41] is a method that dynamically optimizes prompts during the testing phase, enhancing the model's zero-shot generalization ability. Specifically, TPT generates multiple augmented views of the test sample and then minimizes the entropy of the model's output logits across them to ensure consistent prediction. Recently, many works built upon TPT have been proposed to further enhance the performance of vision-language models. Particularly, DiffTPT [6] leverages the power of diffusion models to generate semantically consistent augmented images for entropy minimization. PromptAlign [39] bridges the gap between the test sample and source distribution by aligning token statistics, including mean and variance. Nevertheless, these approaches require gradient descent over the augmented images, which is computationally expensive and time-consuming.

**Training-free Test-time Adaptation**  applies cache model or prototypes to make prediction of test samples in a non-parametric manner [15, 17, 53]. T3A [15] utilizes prototypes as downstream classifiers and dynamically adjusts the weights. AdaNPC [53] leverages the data from the source domain to address the issues of computation overhead and domain forgetting. For vision-language models, TDA [17] introduces both positive cache and negative cache to obtain high-quality test samples from the target domain. However, these methods only consider inter-sample interactions and may fail to generalize well when the downstream tasks require fine-grained knowledge or there is insufficient similarity across samples.

## 3  Methodology

### 3.1  Preliminary

**Problem setting.**  We begin by introducing the basic notations in test-time adaptation. We consider binary classification for simplicity and the theory can be easily extended to multi-classifications settings. Let $p_t(x, y)$ denotes the joint distribution of image and labels in the target distribution, and

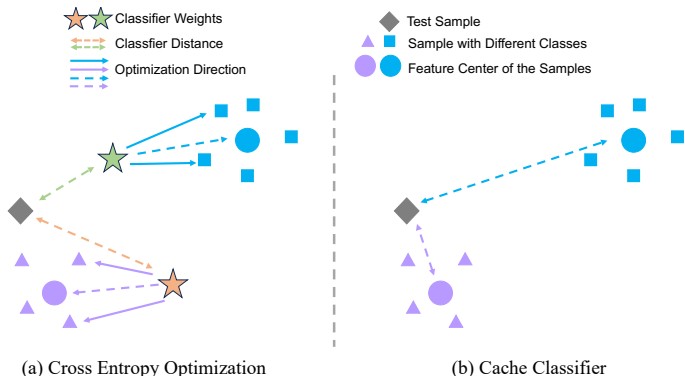



(a) Cross Entropy Optimization        (b) Cache Classifier



Figure 2: Connection between cross-entropy optimization and cache classifier over well-clustered samples with a frozen feature encoder. With optimization of cross-entropy, samples will pull the classifier weights closer of the same class while pushing them away from different class weights. Since the feature space is well-clustered, the classifier weights will ultimately converge near the feature center of the samples. Finally, the optimal classifier achieved through cross-entropy minimization will exhibit similar behavior with the cache classifier.

we simply assume that samples $\{(x_i, y_i)\}_{i=1}^n$ are drawn i.i.d. from the distribution with $y_i$ represents the one-hot label.

**Definition 1.** *(Classification error.) Given $f$ as a binary classification function. The error incurred by hypothesis $f \in \mathcal{H} : \mathcal{X} \to \{0, 1\}$ under the distribution $p_t(x, y)$ can be defined as*

$$\epsilon(f) = \mathbb{E}_{p_t(x,y)}[f(x) \neq y] = \mathbb{E}_{p_t(x,y)}[|f(x) - y|], \tag{1}$$

*the last equality holds in a binary classification setting.*

**Definition 2.** *(Excess error.) Given the Bayes classifier under distribution $p_t(x)$: $f^*(x) = \mathbb{I}\{f(x) \geq 1/2\}$ and the optimal classfier $f^*$, the excess error of $f$ is defined as*

$$\mathcal{E}(f) = \epsilon(f) - \epsilon(f^*) = 2\mathbb{E}_{x \sim p_t(x)}\left[\left|f(x) - \frac{1}{2}\right| \mathbb{I}\{f(x) \neq f^*(x)\}\right] \tag{2}$$

**CLIP classifier** Let $g$ be the image encoder of CLIP, $C$ be the feature dimension, $N$ denotes the number of categories, $w_i \in R^C$ represents the $i_{th}$ text embedding cluster. Considering normalized embedding $w$ and $g(x)$, we can derive a simplified version of the output of CLIP for class $i$:

$$Z_i = w_i^T g(x). \tag{3}$$

And we denote the output logits as $\boldsymbol{p}(x) = [Z_1, Z_2, ..., Z_N] \in R^N$.

**Cache classifier** Given an unseen sample $x$, encoder $g$ with dimensional $C$, cache size $K$ and number of categories $N$, the cache classfier conduct feature retrieval based on the similarity with the data $\{(x_i, y_i)\}_{i=1}^K$ in the cache. The predictions based on Tip-Adapter [52] are as follows:

$$\boldsymbol{p_{cache}}(x) = A\left(g(x)G_{cache}^T\right)Y, \tag{4}$$

where $A(z) = \alpha \exp(-\beta(1-z))$ denotes a scaling function with a weighting factor $\alpha$ and a smoothing scalar $\beta$, $G_{cache} \in R^{K \times C}$ represents the feature of $K$ samples $\{x_i\}_{i=1}^K$ in the cache and $Y \in R^{K \times N}$ is the corresponding labels $\{y_i\}_{i=1}^K$. Considering the number of samples in class $y_i$, We can also derive a simplified version of Eq.(4) as follows, by ignoring the scaling function and adopting an instance-wise computation style:

$$\boldsymbol{p_{cache}}(x) = \sum_{i=1}^k \alpha_i \left[g(x_i)^T g(x)\right] y_i, \tag{5}$$

where $\alpha_i = \frac{1}{n_{y_i}}$ for class balance or $\alpha_i = \frac{1}{\sum_{j=1}^k [g(x_j)^T g(x)]}$ for normalization across all the samples.

## 3.2 A Closer Look at Entropy-based and Cache-based Methods

We start with analyzing the filtering operation of augmented images in TPT. Pseudo-labels tends to be noisy in the test time, and entropy can serve as a confidence metric to identify trustworthy samples

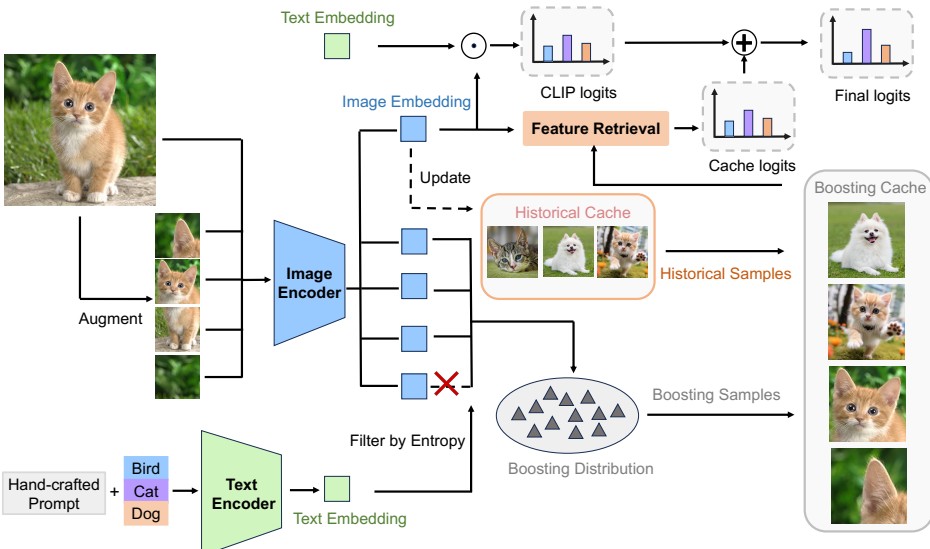

Figure 3: **Overall architecture of BoostAdapter.** BoostAdapter leverages knowledge from the target domain and employs self-bootstrapping with historical and boosting samples in the boosting cache, respectively.

among augmented views [43, 41, 33]. These high-quality samples can be considered drawn i.i.d. from the so-called boosting distribution as defined below.

**Definition 3.** *(**Boosting Distribution**.) Given a test sample from target distribution $x \sim p_t(x)$, let $H(\cdot)$ be the entropy measuring function and $Aug(\cdot)$ be the regional augmentation. By filtering noisy samples based on threshould $\tau$, we have the following property of boosting distribution $p_b(x)$:*

$$\hat{x} \sim p_b(x) \to \{\hat{x} = Aug(x) \wedge H(\boldsymbol{p}(x)) \le \tau\} \tag{6}$$

We also terms the samples from the boosting distribution as **boosting samples**. Then we can connect entropy-based methods and cache classifier by the following proposition:

**Proposition 1.** *(Informal) Given $n$ samples $\{(x_i, y_i)\}_{i=1}^n$ with a freeze encoder $g$ that effectively performing feature clustering with respect to labels, the gradient descent optimization direction of the classifier's weights based on cross-entropy generally tends towards making predictions using the cache classifier with class balance weights defined in 5 on these samples.*

An intuitive illustration of Proposition 1 is depicted in Figure 2, where the weights of optimal classfier behave like the feature centers across different classes with of the well-clusterd samples. Revisiting the entropy-based method TPT, when provided with high-quality boosting samples with low entropy drawn from the boosting distribution, the objective function of entropy minimization optimizes in a manner similar to conducting cross-entropy optimization over the pseudo-labels. According to Proposition 1, TPT performs similarly to the cache-based methods with a cache comprising the same boosting samples from the boosting distribution.

### 3.3 Boosting your Training-free Adapters

Existing cache-based methods store historical test samples only as useful information for prediction. In light of the analysis above, we can integrate the idea behind TPT into these training-free adapters by incorporating boosting samples into the memory bank. In particular, each sample can participate in both inter-sample and intra-sample interactions with the instance-agnostic historical samples and the instance-aware boosting samples in the cache, respectively.

Specifically, with $k_t$ selected historical samples and $k_b$ selected boosting samples to comprise the cache, we extend the classifier defined in Eq.(4) and formulate our BoostAdapter as follows:

$$\boldsymbol{p_{boost}}(x) = A\left(g(x)\tilde{G}_{cache}^T\right)\tilde{Y}, \tag{7}$$

where $A$ is the same scaling function defined in Eq.(4), $\tilde{G}_{cache} \in R^{(k_t+k_b)\times C}$ denotes the features of the combination of both the historical and boosting samples, and $\tilde{Y} \in R^{(k_t+k_b)\times N}$ is the label.

Since we do not have access to the labels of the test samples, we generate one-hot pseudo-labels for them using argmax operations. However, these pseudo-labels tend to be noisy in the target domain. Therefore, we apply filtering based on entropy thresholds on the test data stream following [41] to obtain trustworthy historical samples. We employ a similar operation to select boosting samples from multiple augmented views of the current sample. In practice, we dynamically adapt the entropy thresholds $\tau$ for each test sample, with a fixed percentile $p$. The cache continuously updates with lower entropy historical samples from the test data stream, while the current test sample augments the cache with self-boosting samples and forms an independent cache that only affects its own prediction. Additionally, to maintain diversity while considering the relevance to each test sample, we set a maximum shot capacity for each class $k$ in the cache. This means that samples in the cache will be replaced by a lower-entropy historical sample or boosting sample when necessary.

An important issue is whether introducing boosting samples brings improvements to the training-free adapters. We will first make some necessary assumptions and then theoretically verify the effectiveness in reducing target error by incorporating samples from the boosting distribution.

**Assumption 1.** *(Strong Density Condition) For any test sample $x_0$ in the target distribution $x_0 \sim p_t(x)$ and the boosting distribution $p_b(x_0)$, given positive lower bound $m$ and upper bound $M$, positive scaling constant $c_t$ and $c_b$, the radius bound $R > 0$, and $\mathcal{B}(x, r) = \{x' : \| x' - x \| \le r\}$ is the ball centered on $x$ with radius $r$. We assume $p_t(x)$ and $p_b(x_0)$ are absolutely continuous with respect to the Lebesgue measure in $\mathbb{R}^d$. For $r \in (0, R]$, we assume*

$$\begin{cases} \lambda[p_t(x) \cap \mathcal{B}(x_0, r)] \ge c_t \lambda[\mathcal{B}(x_0, r)] \\ \lambda[p_b(x_0) \cap \mathcal{B}(x_0, r)] \ge c_b \lambda[\mathcal{B}(x_0, r)] \\ m < \dfrac{dp_t(x)}{d\lambda} < M; m < \dfrac{dp_b(x)}{d\lambda} < M, \end{cases} \tag{8}$$

*where $\lambda$ is the Lebesgue measure in Euclidean space.*

**Assumption 2.** *(L-Lipschitz Condition) Let $f$ be the classification function and $L$ be a positive constant. For all feasible $x, x'$ we have $|f(x) - f(x')| \le L \| x - x' \|$.*

**Assumption 3.** *(Low Noise Condition). Let $\beta, C_\beta$ be positive constants and we assume $p_t(x)$ satisfies $P_{x \sim p_t(x)} \left( \left| f(x) - \frac{1}{2} \right| < t \right) \le C_\beta t^\beta$ for all $t > 0$.*

**Remark** Assumption 1 intuitively ensures that for any test sample, there is a surrounding neighborhood with a significant presence of samples from the target domain and the boosting distribution. More importantly, for a specific sample $x_0$, boosting samples $x \sim p_b(x_0)$ should be closer to $x_0$ than other samples $x \sim p_t(x)$ from the target domain, *i.e.*, generally, we have $c_t \le c_b$. Assumption 2 and 3 describe the smoothness of functions and imply a high level of confidence in predictions around the threshold, respectively.

**Proposition 2.** *(Historical Cache reduce Emperical Risk) Given $f$ as the training-free classfier consisting of historical samples only defined by Eq.(4). Let $n_t$ to be the number of confident previously predicted samples in the target domain and $k_t$ as the number of historical samples in the cache, with assumptions 1-3, the following results hold with high-probability for large enough $k_t$ and $n_t$.*

$$\mathcal{E}(f) \le \mathcal{O}\left( \left( \left( \frac{1}{k_t} \right)^{1/4} + \left( \frac{k_t}{c_t n_t} \right)^{1/d} \right)^{1+\beta} \right) \tag{9}$$

**Proposition 3.** *(Historical Cache benefits from Boosting Samples) Let $n_t$ to be all confident previously predicted samples in the target domain and $n_b$ be the number of boosting samples that are drawn from the boosting distribution. Given $k_t$ and $k_b$ to be the number of historical samples and the number of boosting samples to be selected as the nearest neighbors stored in the cache, respectively. Let $w_{ti}$ and $w_{bi}$ be the weights defined in Eq.(5) of the historical samples and boosting samples. We have the following bound for the empirical risk of the cache classfier defined in 7.*

$$\mathcal{E}(f) \le \mathcal{O}\left( \left( \left( \frac{1}{k_t + k_b} \right)^{1/4} + \sum_{i=1}^{k_t} w_{ti} \left( \frac{k_t}{c_t n_t} \right)^{1/d} + \sum_{i=1}^{k_b} w_{bi} \left( \frac{k_b}{c_b n_b} \right)^{1/d} \right)^{1+\beta} \right). \tag{10}$$

Table 1: **Full results on the OOD benchmark with ViT-B/16 backbone.** We report top-1 accuracy and "Average" is calculated by taking the mean accuracy across all four OOD datasets.

| | Imagenet-V2 | Imagenet-Sketch | Imagenet-A | Imagenet-R | Average |
|---|---|---|---|---|---|
| CLIP [36] | 60.86 | 46.09 | 47.87 | 73.98 | 57.20 |
| CLIP+TPT [41] | 64.35 | 47.94 | 54.77 | 77.06 | 60.81 |
| CoOp [55] | 64.20 | 47.99 | 49.71 | 75.21 | 59.28 |
| CoOp+TPT [41] | **66.83** | 49.29 | 57.95 | 77.27 | 62.84 |
| Co-CoOp [54] | 64.07 | 48.75 | 50.63 | 76.18 | 59.91 |
| Co-CoOp+TPT [41] | 64.85 | 48.27 | 58.47 | 78.65 | 62.61 |
| Maple [18] | 64.07 | 49.15 | 50.90 | 76.98 | 60.28 |
| Maple + TPT [41] | 64.87 | 48.16 | 58.08 | 78.12 | 62.31 |
| PromptAlign [39] | 65.29 | 50.23 | 59.37 | 79.33 | 63.55 |
| DiffTPT [6] | 65.10 | 46.80 | 55.68 | 75.00 | 60.52 |
| TDA [17] | 64.67 | 50.54 | 60.11 | 80.24 | 63.89 |
| BoostAdapter | 65.51 | **51.28** | **64.53** | **80.95** | **65.57** |

Table 2: **Full results on the Cross-Domain Benchmark with ViT-B/16 backbone.** We report top-1 accuracy and "Average" is calculated by taking the mean accuracy across all ten datasets. The error bound is $\pm 0.17$.

| | Caltech | Pets | Cars | Flowers | Food101 | Aircraft | SUN397 | DTD | EuroSAT | UCF101 | Average |
|---|---|---|---|---|---|---|---|---|---|---|---|
| CLIP [36] | 93.35 | 88.25 | 65.48 | 67.44 | 83.65 | 23.67 | 62.59 | 44.27 | 42.01 | 65.13 | 63.58 |
| CLIP+TPT [41] | 94.16 | 87.79 | 66.87 | 68.98 | 84.67 | 24.78 | 65.50 | **47.75** | 42.44 | 68.04 | 65.10 |
| CoOp [55] | 93.70 | 89.14 | 64.51 | 68.71 | 85.30 | 18.47 | 64.15 | 41.92 | 46.39 | 66.55 | 63.88 |
| CoCoOp [54] | 93.79 | 90.46 | 64.90 | 70.85 | 83.97 | 22.29 | 66.89 | 45.45 | 39.23 | 68.44 | 64.63 |
| MaPLe [18] | 93.53 | 90.49 | 65.57 | 72.23 | 86.20 | 24.74 | 67.01 | 46.49 | 48.06 | 68.69 | 66.30 |
| MaPLe+TPT [41] | 93.59 | 90.72 | 66.50 | 72.37 | 86.64 | 24.70 | 67.54 | 45.87 | 47.80 | 69.19 | 66.50 |
| DiffTPT [6] | 92.49 | 88.22 | 67.01 | 70.10 | 87.23 | 25.60 | 65.74 | 47.00 | 43.13 | 62.67 | 65.47 |
| PromptAlign [39] | 94.01 | **90.76** | 68.50 | **72.39** | 86.65 | 24.80 | 67.54 | 47.24 | 47.86 | 69.47 | 66.92 |
| TDA [17] | 94.24 | 88.63 | 67.28 | 71.42 | 86.14 | 23.91 | 67.62 | 47.40 | 58.00 | 70.66 | 67.53 |
| BoostAdapter | **94.77** | 89.51 | **69.30** | 71.66 | **87.17** | **27.45** | **68.09** | 45.69 | **61.22** | **71.93** | **68.68** |

**Remark** Proposition 2 provides a guarantee of the effectiveness of selecting $k_t$ out of $n_t$ historical samples to comprise the cache. The empirical risk is quite small when $n_t \to \infty$ since the cache captures the full information of the target domain. Proposition 3 demonstrates that the historical cache can further reduce empirical risk by incorporating $k_b$ boosting samples.

## 4 Experiments

### 4.1 Experimental Setup

**Datasets** Following the setting in TPT [41], we conduct experiments on both Out-of-Distribution (OOD) benchmark and Cross-Domain benchmark. The OOD benchmark evaluates the model's robustness to natural distribution shifts on 4 ImageNet [4] Variants, including ImageNetV2 [37], ImageNet-Sketch [44], ImageNet-A [14] and ImageNet-R [13]. We evaluate the transferring performance on 11 datasets in the Cross-Domain benchmark: Aircraft [31], Caltech101 [5], Cars [19], DTD [3], EuroSAT [11], Flower102 [32], Food101 [2], Pets [34], SUN397 [48],and UCF101 [42]. We follow the split in [55] and report the top-1 accuracy. The error bound are also provided.

**Implementation details** We utilize a pre-trained ViT-B/16 of CLIP as the foundation model. In test-time adaptation, the batch size is set to be 1. We search for the optimal shot capacity to balance diversity and relevance of samples. For boosting samples, we utilize random crop and then random horizontal flip as augmentations. Moreover, we empirically set the entropy threshold percentile to $p = 0.1$ and filter 64 augmented views based on random cropping to obtain the boosting samples. and filter 64 augmented views to obtain the boosting samples. The top-1 accuracy and the error bound is reported on the test sets. All our experiments are conducted with a Nvidia 3090 24GB GPU.

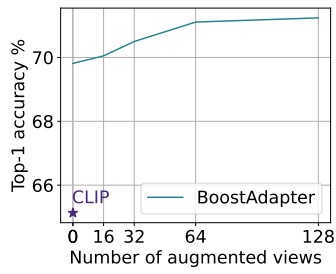

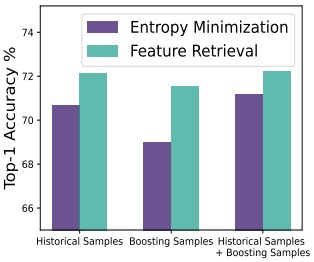

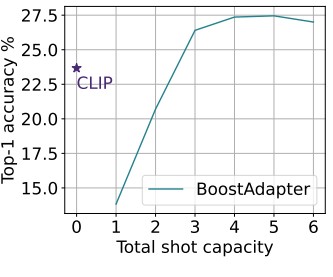

| (a) Number of augmented views | (b) Adaptation methods | (c) Total shot capacity |

Figure 4: Ablation studies of (a) number of augmented views to generate boosting samples (b) different adaptation methods and (c) total shot capacity of the cache.

Table 3: **Ablation study on historical samples and boosting samples on the OOD benchmark with ViT-B/16 backbone.** We report top-1 accuracy and the error bound is ±0.12.

|  | -V2 | -Sketch | -A | -R | Average |
|---|---|---|---|---|---|
| CLIP | 60.86 | 46.09 | 47.87 | 73.98 | 57.20 |
| Historical Samples | 64.93 | 50.23 | 63.80 | 80.43 | 64.85 |
| Boosting Samples | 65.40 | 50.59 | 64.40 | **80.96** | 65.34 |
| BoostAdapter | **65.51** | **51.28** | **64.53** | 80.95 | **65.57** |

Table 4: **Full results on the OOD benchmark with RN-50 backbone.** We report top-1 accuracy and the error bound is ±0.06.

|  | -V2 | -Sketch | -A | -R | Average |
|---|---|---|---|---|---|
| CLIP [36] | 51.41 | 33.37 | 21.83 | 56.15 | 40.69 |
| TPT [41] | 54.70 | 35.09 | 26.67 | 59.11 | 43.89 |
| CALIP [10] | 53.70 | 35.61 | 23.96 | 60.81 | 43.52 |
| CoOp [55] | 55.40 | 34.67 | 23.06 | 56.60 | 42.43 |
| CoCoOp [54] | 55.72 | 34.48 | 23.32 | 57.74 | 42.82 |
| DiffTPT [6] | 55.80 | 37.10 | 31.06 | 58.80 | 45.69 |
| TDA [17] | 55.54 | 38.12 | 30.29 | 62.58 | 46.63 |
| BoostAdapter | **56.14** | **38.87** | **35.12** | **62.66** | **48.20** |

## 4.2 Out-of-Distribution Generalization

To verify the robustness of BoostAdapter, we evaluate our method on the OOD benchmark, in comparison with existing training-require methods including CoOp [55], CoCoOp [54], TPT [41], DiffTPT [6], Maple [18] and PromptAlign [39], as well as training-free method TDA [17]. As can be seen from Table 8, the most striking observation emerging from the comparison is that BoostAdapter significantly outperforms other baselines on average and improves the generalization ability of the model. For training-free methods such as TPT, DiffTPT and PromptAlign, BoostAdapter achieves superior performance while saving on optimization computation overhead. For training-free methods like TDA, BoostAdapter gains consistent performance improvements with the introduction of the boosting samples. Notably, BoostAdapter surpasses TDA by 4.42% on ImageNet-A and 0.84% on ImageNet-V2, respectively. This enhancement indicates the effectiveness of self-bootstrapping when historical samples may not provide sufficient useful information.

## 4.3 Cross-Domain Transfer

We further highlight our improvements in the transfer ability of CLIP on the Cross-Domain benchmark and present the results in Table 2. Compared with existing training-required and training-free methods, BoostAdapter achieves state-of-the-art performance on 7 out of 10 tasks, surpassing the strongest baselines by an average of 1.15%. With diverse classes at test time, regional boosting enables BoostAdapter to adaptively extract knowledge that makes classes distinct from each other in a multi-scale manner. Notably, for datasets requiring fine-grained information for classification such as Aircraft, the improvement of BoostAdapter is most significant.

## 4.4 Ablation Study

**Historical Samples and Boosting Samples.** To demonstrate the effect of historical and boosting samples, we introduce two variants of BoostAdapter that utilize only historical samples or only boosting samples, respectively. Additionally, we provide the zero-shot results of CLIP for comparison. As shown in Table 3, CLIP significantly benefits from both historical samples and boosting samples, resulting in notable improvements in performance. The consistent improvement of BoostAdapter compared to the variant that utilizes only historical samples further confirms the effectiveness of

Table 5: **Full results on the Cross-Domain Benchmark with RN-50 backbone.** We report top-1 accuracy and "Average" is calculated by taking the mean accuracy across all ten datasets. The error bound is ±0.05.

| | Caltech | Pets | Cars | Flowers | Food101 | Aircraft | SUN397 | DTD | EuroSAT | UCF101 | Average |
|---|---|---|---|---|---|---|---|---|---|---|---|
| CLIP [36] | 85.88 | 83.57 | 55.70 | 61.75 | 73.97 | 15.66 | 58.8 | 40.37 | 23.69 | 58.84 | 55.82 |
| CLIP + TPT [41] | 87.02 | 84.49 | 58.46 | 62.69 | 74.88 | 17.58 | 61.46 | 40.84 | 28.33 | 60.82 | 57.66 |
| CALIP [10] | 87.71 | 86.21 | 56.27 | 66.38 | 77.42 | 17.76 | 58.59 | 42.39 | 38.90 | 61.72 | 59.34 |
| DiffTPT [6] | 86.89 | 83.40 | **60.71** | 63.53 | **79.21** | 17.60 | 62.72 | 40.72 | 41.04 | 62.67 | 59.85 |
| CuPL [35] | 89.29 | 84.84 | 57.28 | 65.44 | 76.94 | **19.59** | 62.55 | **48.64** | 38.38 | 58.97 | 60.19 |
| TDA [17] | **89.70** | **86.18** | 57.78 | **68.74** | 77.75 | 17.61 | 62.53 | 43.74 | **42.11** | 64.18 | 61.03 |
| BoostAdapter | 88.48 | 85.75 | 59.67 | 68.25 | 78.78 | 18.93 | **62.83** | 43.85 | **44.40** | **64.42** | **61.54** |

Table 6: **Comparisons with baselines on ImageNet-C at severity level 5 regarding accuracy (%).**

| | Noise | | | Blur | | | | Weather | | | | Digital | | | | |
|---|---|---|---|---|---|---|---|---|---|---|---|---|---|---|---|---|
| | Gauss. | Shot | Impul. | Defoc. | Glass | Motion | Zoom | Snow | Frost | Fog | Brit. | Contr. | Elastic | Pixel | JPEG | Avg. |
| CLIP-ViT-B/16 | 15.15 | 16.28 | 15.26 | 25.83 | 16.87 | 26.34 | 24.43 | 34.56 | 33.01 | 39.10 | 57.78 | 18.45 | 14.71 | 35.62 | 35.81 | 27.28 |
| TDA | 17.50 | 18.59 | 18.12 | 59.12 | 19.02 | 28.25 | 26.24 | 37.30 | 35.30 | 41.57 | 59.04 | 21.06 | 17.61 | 37.78 | 37.26 | 31.58 |
| BoostAdapter | **17.53** | **18.89** | **18.39** | **59.70** | **19.07** | **28.62** | **27.33** | **38.21** | **36.13** | **42.31** | **59.63** | **21.22** | **18.23** | **39.25** | **38.07** | **32.17** |

incorporating boosting samples into the training-free adapters. See Section E in the Appendix for more results.

**Number of Augmented Views for Boosting Samples.** We augment the testing samples and filter them by mutual information with the CLIP text embedding to obtain the boosting samples. We vary the number of augmented views and investigate the performance of BoostAdapter on UCF101 in Figure 4a. With a larger number of augmented views, the performance improves due to more bootstrapping information of the test sample, which is consistent with the conclusions of TPT [41] and PromptAlign [39]. However, the computational overhead also increases with more augmented views, and selecting 64 augmented views is a fair trade-off between boosting performance and efficiency.

**Adaptation Methods.** Training-required methods use entropy as a self-supervised objective, whereas training-free methods classify samples based on feature retrieval. We compare the performance of these two adaptation methods under the constraints of historical samples only, boosting samples only, or both, and present the results on Flower102 in Fig. 4b. Entropy minimization requires gradient descent and model optimization, resulting in high training costs and relatively lower performance across all three settings. In contrast, the training-free methods based on feature retrieval offer significant performance improvements with lower computational overhead. Additionally, both adaptation methods benefit from combining historical samples and boosting samples, consistent with the conclusions in Table 3.

**Total shot capacity.** BoostAdapter maintains low-entropy samples per class in the cache, and Figure 4c studies the influence of different total shot capacities containing historical samples and boosting samples of each class on Aircraft. As can be observed from the results, when the cache capacity is small, the low-entropy samples maintained by BoostAdapter do not necessarily provide a benefit for classification compared to CLIP. As the shot capacity increases, BoostAdapter will achieve the best balance of diversity and relevance, and a larger capacity does not guarantee better performance.

**Versatility.** To demonstrate the versatility of BoostAdapter, we apply it to the RN-50 backbone and present the results in Tables 4 and 5. The improvement is consistent and on average, BoostAdapter outperforms TDA by 1.57% on the OOD benchmark and 0.49% on the Cross-Domain benchmark.

## 4.5 Discussions

**Generalization on Corruption Datasets** To further evaluate the generalization ability of Boost-Adapter in new test-time scenarios, we compare BoostAdapter with baseline methods on the Imagenet-C dataset at the highest severity level 5. The key observation from Table 6 is that BoostAdapter

Table 7: **Efficiency analysis.** We evaluate different methods on a single NVIDIA 3090 24GB GPU and report the frames per second (fps) and memory cost (GB).

| | Augmentation | Views | Inference Speed (fps) | Memory (GB) | OOD Results | Cross-Domain Results |
|---|---|---|---|---|---|---|
| CLIP | - | - | 82.3 | 0.7 | 57.20 | 63.58 |
| TPT | Augmix | 64 | 0.29 | 4.5 | 60.81 | 65.10 |
| DiffTPT | Diffusion | 64 | 0.10 | 14.4 | 60.52 | 66.92 |
| TDA | Augmix | 64 | 11.89 | 1.2 | 63.89 | 67.53 |
| BoostAdapter | Rand. Crop & Rand. Horiz. Flip | 64 | 11.23 | 1.2 | 65.57 | 68.68 |

Table 8: **Unification of more training-required methods.** BoostAdapter benefits from different training-required methods.

| | -V | -S | -A | -R | Average |
|---|---|---|---|---|---|
| CLIP-ViT-B/16 | 60.86 | 46.09 | 47.87 | 73.98 | 57.20 |
| TDA | 64.67 | 50.54 | 60.11 | 80.24 | 63.89 |
| BoostAdapter | 65.51 | 51.28 | 64.53 | 80.95 | 65.57 |
| BoostAdapter+ TSD | 65.49 | 51.50 | 64.37 | 81.15 | 65.63 |
| BoostAdapter+ DEYO | 65.71 | 51.52 | 64.65 | 81.43 | 65.83 |

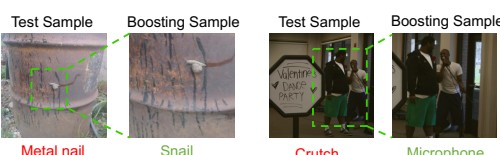

Figure 5: **Qualitative results.** The model predictions are provided below the images. Boosting samples with low entropy improves information extraction from the test sample and helps the model to distinguish better.

consistently outperforms TDA across all 15 corruption types, highlighting its practical applicability in real-world situations. The superior performance of BoostAdapter stems from its capability to capture the knowledge of the test sample even under severe corruption. This is achieved with the help of the boosting samples, which effectively filter out noisy parts while retaining useful information.

**Efficiency Analysis** BoostAdapter requires augmentation over the test samples, which may slightly affect the inference speed during testing. We conduct an efficiency analysis of BoostAdapter in comparison with existing Test Time Augmentation (TTA) methods and provide the results in Table 7. BoostAdapter is slightly slower than the cache-based method TDA, yet still significantly faster than training-required methods. The memory cost of BoostAdapter is also comparable to other baselines.

**Unification of Training-required and Training-free Methods.** From the unified perspective, we can also enhance training-free adapters with additional training-required methods. Here we take TSD [45] and DEYO [21] as the showcase. Specifically, in the BoostAdapter+DEYO variant, we filter out augmented views with a PLPD lower than 0.2. For the BoostAdapter TSD variant, we discard augmented views that have different cache predictions and CLIP predictions to ensure consistency of the boosting samples. When equipping BoostAdapter with the technique of TSD and DEYO, we observe further improvement and find that training-free adapters can benefit from various boosting techniques of training-required methods.

**Qualitative Results** The qualitative results are provided in Figure. 5. By incorporating samples with low entropy from regional bootstrapping, the model is enhanced to more effectively capture the fine-grained information of the test samples, thereby improving the overall performance.

# 5 Conclusions

In this work, we present an insightful analysis of existing training-required and training-free TTA methods to bridge the gap between them. In particular, we improve training-free adapters by incorporating self-boosting samples into the memory bank inspired by the idea of regional bootstrapping from entropy-based methods. The cache in our method, containing instance-agnostic historical samples and instance-aware boosting samples, is capable of performing knowledge mining on both the target domain and the testing sample itself. We also derive error bounds in the test-time adaptation setting and show that this cache benefits from both historical samples and boosting samples. Extensive experiments on the two benchmarks demonstrate the effectiveness of our method.

Despite the promising performance of our method, it also has some limitations. It requires slightly more computation overhead than existing training-free adapters due to the multiple augmentation of the test samples, as discussed in Appendix. One future direction is to develop a more efficient augmentation method to obtain boosting samples, rather than merely randomly cropping and then filtering over the test samples.

## Acknowledgements

This work is supported in part by the National Natural Science Foundation of China, under Grants (624B2088, 62302309, 62171248), Shenzhen Science and Technology Program (JCYJ20220818101014030, JCYJ20220818101012025), and PCNL KEY project (PCL2023AS6-1).

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

# Appendix

## A Dataset and Licenses

Table 9 presents the statistics and details of datasets used in the paper. We also provide the corresponding license information of the datasets and source code.

**Datasets.** Below are the datasets used in this paper that have known license information: The following datasets used in this paper are under the MIT License: ImageNet-A [14], ImageNet-V2 [37], ImageNet-R [13], ImageNet-Sketch [44], EuroSAT [11], Food101 [2].
The following datasets used in this paper are under the CC BY-SA 4.0 License: Oxford-Pets [34], Caltech101 [5].
The following datasets used in this paper are for research purposes only: DTD [3], StanfordCars [19], SUN397 [48], FGVC-Aircraft [31], Flower102 [34], UCF101 [42].

**Source code.** We use the implementation of existing baseline methods for reporting their results in this paper. Below are their license information: Source code used in this paper that are under the MIT License: CLIP [36], PromptAlign [39] and TDA [17].

| Dataset | Description | Classes | Test Size |
|---|---|---|---|
| | Out-of-Distribution Benchmark | | |
| ImageNet-V2 | New Validation Sets of ImageNet | 1,000 | 10,000 |
| ImageNet-S | Sketch Images | 1,000 | 50,000 |
| ImageNet-A | Natural Adversarial Examples | 200 | 7,500 |
| ImageNet-R | Rendition Extension of ImageNet | 200 | 30,000 |
| | Cross-Domain Benchmark | | |
| Aircraft | Aircraft Model Classification | 100 | 3,333 |
| Caltech101 | Natural Image Classification | 100 | 2,465 |
| Cars | Cars Classification | 196 | 8,041 |
| DTD | Describable Textures Dataset | 47 | 1,692 |
| EuroSAT | Satellite Images | 10 | 8,100 |
| Flowers102 | Flowers Classification | 102 | 2,463 |
| Food101 | Food Classification | 101 | 30,300 |
| Pets | Pets Classification | 37 | 3,669 |
| SUN397 | Scene Categorization Benchmark | 397 | 19,850 |
| UCF101 | Action Recognition Dataset | 101 | 3,783 |

Table 9: Datasets statistics.

## B Broader Impacts

In this paper, we focus on bridging the gap between training-required and training-free methods to improve the generalization ability of vision-language models. We also theoretically derive the error bound of incorporating boosting samples into the historical cache. We hope that our work will inspire the community to explore test-time adaptation in an effective and efficient way.

## C Theoretical Proof

### C.1 Cross-entropy Optimization behaves like Cache Classifier over well-clustered Samples (Proof of Proposition 1)

Given well-clustered samples in the feature space and the classifier defined in Eq.(3), we first derive the distance between the weights of the classifier and the optimal weights and then establish the connection between the optimal weights with the features center of the samples.

Suppose the classifier function $f$ over samples is convex and differentiable, and also $L$-smooth. Let the distance between initial weight $w^{(0)}$ and optimal weight $w^*$ to be $D = ||w^{(0)} - w^*||$. GD updates by $w^{(t+1)} = w^t - f_t^* \nabla f(w^t)$ with step size $f_t^* = \frac{1}{L}$, and then GD enjoys the following convergence guarantee:

$$||w - w^*|| \leq \frac{2L \left\| w^{(0)} - w^* \right\|^2}{T - 1} = \mathcal{O}\left(\frac{LD^2}{T}\right). \tag{11}$$

We then showcase the relationship between $w_*$ and the features center $\mu_i$ of class $i, i = 1, 2, ..., N$. Since we optimize on well-clustered samples, we consider the scenarios of perfect clusters, where samples in the class $i$ will be encoded into the same point $\mu_i$ by the encoder $g$, and these points should be farthest enough between each other. Given $n$ samples $\{(x_k, y_k)\}_{k=1}^n$, with the number of samples in class $i$ to be $n_i$, the cross-entropy loss function $L$ can be written as:

$$L = -\sum_{i=1}^n \log P(y = y_k | x_k) \tag{12}$$

Substitute the sample $g(x_k) = \mu_i$ from class $i$, we derive the probability $P(y = i|x_k)$ using the softmax function from Eq.(3) is:

$$P(y = i|x_k) = \frac{exp(w_i^T \mu_i)}{\sum_{j=1}^N exp(w_j^T \mu_i)}. \tag{13}$$

Thus, the cross-entropy loss for a sample $(x_k, y_k = i)$ is:

$$L_k = -\log\left(\frac{exp(w_i^T \mu_i)}{\sum_{j=1}^N exp(w_j^T \mu_i)}\right). \tag{14}$$

For all samples, the total loss is:

$$L = -\sum_{i=1}^N n_i \log\left(\frac{exp(w_i^T \mu_i)}{\sum_{j=1}^N exp(w_j^T \mu_i)}\right). \tag{15}$$

The gradient of the loss with respect to $w_i$ can be simplified as:

$$\frac{\partial L}{\partial w_i} = -\mu_i n_i + \mu_i \sum_{k=1}^N n_k \frac{exp(w_i^T \mu_k)}{\sum_{j=1}^N exp(w_j^T \mu_k)}. \tag{16}$$

When converges to the optimal weight, we have the condition of fixed point $\frac{\partial L}{\partial w_i^*} = 0$. And we have

$$-\mu_i n_i + \mu_i \sum_{k=1}^N n_k \frac{exp((w_i^*)^T \mu_k)}{\sum_{j=1}^N exp((w_j^*)^T \mu_k)} = 0. \tag{17}$$

Thus, we have

$$\sum_{k=1}^N n_k \frac{exp((w_i^*)^T \mu_k)}{\sum_{j=1}^N exp((w_j^*)^T \mu_k)} = n_i. \tag{18}$$

Given a well-clustered samples, we could have $exp((w_i^*)^T \mu_k) \gg exp((w_j^*)^T \mu_k)$ for a specific $i$ when $w_i^*$ is near $\mu_k$. Then since the equality in Eq.(18) will hold for each class and for class $i = 1, 2, ..., N$ we have

$$w_i^* \to \mu_i. \tag{19}$$

Combining Eq.(11) and Eq.(18), with iteration steps $T$, we show that the weight of classfier will finally converge to the feature center of each class:

$$||w - \mu|| \leq ||w - w^*|| + ||w^* - \mu|| \leq \mathcal{O}\left(\frac{LD^2}{T}\right). \tag{20}$$

And we have the output logits of the optimal weights with the encoder $g$:

$$\boldsymbol{p}_{cross}(x) = [\mu_1^T g(x), \mu_2^T g(x), ..., \mu_N^T g(x)] \tag{21}$$

Next we discess the behavior of the cache classifier over these samples. Given the number of well-clustered samples in class $i$ to be $n_i$, the output logits of the cache classifier defined in Eq.(5) using samples $\{(x_k, y_k)\}_{k=1}^n$ can be described as follows:

$$\begin{aligned} \boldsymbol{p}_{cache}(x) &= \sum_{k=1}^n \frac{1}{n_{y_i}} [g(x_k)^T g(x)] y_k \\ &= \sum_{i=1}^N \frac{n_i}{n_i} [\mu_i^T g(x)] y_i \\ &= [\mu_1^T g(x), \mu_2^T g(x), ..., \mu_N^T g(x)] \end{aligned} \tag{22}$$

Combining Eq.(21) and Eq.(22), we draw the conclusion that cross-entropy optimization behaves like cache classifier over well-clustered samples.

## C.2 Historical Cache reduce Empirical Risk (Proof of Proposition 2)

We follow the proofs in [53] and extend the conclusion to boosting samples.

### C.2.1 Additional Definitions and Assumptions

**Definition 4.** *(**Wasserstein-distance and the dual form**). Wasserstein distance measures the distance between two probability distributions on a given metric space. It is defined using the concept of optimal transport. For two distributions $\mathbb{P}, \mathbb{Q}$, The $\rho$-th Wasserstein distance is defined as*

$$W_p(\mathbb{P}, \mathbb{Q}) = \left( \inf_{\gamma \in \Pi(\mathbb{P}, \mathbb{Q})} \int_{X \times X} d(x, y)^p d\gamma(x, y) \right)^{1/p} \tag{23}$$

*Here, $\Pi(\mathbb{P}, \mathbb{Q})$ denotes the set of all couplings (or transport plans) $\gamma$ of $\mathbb{P}$ and $\mathbb{Q}$, i.e., joint distributions on $X \times X$ with marginals $\mathbb{P}$ and $\mathbb{Q}$. The idea is to find the optimal way to transport the mass from one distribution to the other with the minimal cost, where the cost is given by the $p$-th power of the distance.*

*The first Wasserstein distance, $W_1(\mathbb{P}, \mathbb{Q})$, often referred to as the Earth-Mover Distance(EMD), has a particularly elegant dual representation. The dual form of $W_1$ leverages the Kantorovich-Rubinstein duality and can be expressed as:*

$$W_1(\mathbb{P}, \mathbb{Q}) = \sup_{\|f\|_{\mathrm{Lip}} \leq 1} \left( \int_X f \, d\mathbb{P} - \int_X f \, d\mathbb{Q} \right) \tag{24}$$

*Here, the supremum is taken over all 1-Lipschitz functions $f$, which are functions satisfying $|f(x) - f(y)| \leq d(x, y)$ for all $x, y \in X$. This representation shows that $W_1$ can be seen as the maximum difference in expected values of a 1-Lipschitz function over the two distributions. In the following part, Wasserstein distance represents the first Wasserstein distance for simplicity and we utilize $W(\cdot, \cdot)$ instead of $W_1(\cdot, \cdot)$.*

Given the definition of the Wasserstein distance, we have the following proposition that derive the empirical risk on the target domain according to Theorem 1 from [40].

**Proposition 4.** *Given two distributions $\mathbb{P}, \mathbb{Q}$, denote $f^* = \arg\min_{f \in \mathcal{H}}(\epsilon_P(f) + \epsilon_Q(f))$ and $\xi = \epsilon_P(f^*) + \epsilon_Q(f^*)$. Assume all hypotheses $h$ are $L$-Lipschitz continuous, the risk of hypothesis $\hat{f}$ is then bounded by*

$$\epsilon_Q(\hat{f}) \leq \xi + \epsilon_P(\hat{f}) + 2L\mathcal{W}(\mathbb{P}, \mathbb{Q}). \tag{25}$$

### C.2.2 Distance between the Ball Distribution with the Target Distribution

When using the cache classifier with historical samples, a large number of samples that are not similar enough from the target domain will be filtered and the selected samples with high weight are all close to the target data. Thus we extend the conclusion in [53] to the distance between the ball distribution with the target distribution. Considering a test sample from the target distribution $x_t \in p_t(x)$ and a distribution consisting of ball center of all the test samples $\Omega := \bigcup_{x_t \in p_t(x)} \mathcal{B}(x_t, r)$, informally, according to Eq.(23), we have the distance between the ball distribution with the target distribution as follows:

$$\mathcal{W}(\Omega, p_t(x)) = \inf_{\gamma \in \Pi[\Omega, p_t(x)]} \iint \| x_t - x_{ball} \| d\gamma(x_t, x_{ball}), \tag{26}$$

where for each $x_{ball} \in \Omega$, we can find at least one $x_t \in p_t(x)$ such that $\| x_{ball} - x_t \| \leq r$, the overall distance will then be bounded by $r$. Specifically, we can choose a density function $\gamma^*$ where $\gamma^*(x_{ball}, x_t) > 0$ only if $\| x_{ball} - x_t \| \leq r$ otherwise 0, then we have

$$\mathcal{W}(\Omega, p_t(x)) = \inf_{\gamma \in \Pi[\Omega, p_t(x)]} \iint \| x_{ball} - x_t \| d\gamma(x_{ball}, x_t)$$
$$\leq \iint \| x_{ball} - x_t \| \gamma^*(x_{ball}, x_t) dx_{ball} x_t \leq r. \tag{27}$$

However, there is no guarantee that each data $x_t \in p_t(x)$ can find a neighbor $\mathcal{B}(x_t, r)$ with $|\mathcal{B}(x_t, r)| > 0$ with all the small $r$. We then provide the probability that the set of neighbors $\mathcal{B}(x_t, r)$ of each $x_t \in p_t(x)$ is not measuring zero with respect to the radius $r$.

As defined in the cache classfier Eq.(5), we denote $k_t$ is the number of historical samples we select in the cache and $n_t$ is the total number of data from the historical stream. With the strong density assumption, given the coefficient bound $m$ and $M$, for any $x_t \in p_t(x), r < R$, according to Assumption 1, we have

$$|\hat{x}_t \in p_t(x) \wedge \hat{x}_t \in \mathcal{B}(x_t, r)| = \int_{\mathcal{B}(x_t, r) \cap p_t(x)} \frac{dp_t(x)}{d\lambda}(\hat{x}_t) d\hat{x}_t$$
$$\geq m\lambda(\mathcal{B}(x_t, r) \cap p_t(x))$$
$$\geq mc_t \pi_d r^d, \tag{28}$$

where $\pi_d = \lambda(\mathcal{B}(0, 1))$ is the volume of the $d$ dimension unit ball and $\lambda$ is the Lebesgue measure of a set in a Euclidean space. Set $r_0 = (\frac{2k}{mc_t \pi_d n_t})^{1/d}$, with a additional assumption that we utilize a small $k_t$ compared to $n_t$ so that $\frac{k_t}{n_t} < \frac{c_t m \pi_d r_\mu^d}{2}$, we have $r_0 < R$. Then for any $x_t \in p_t(x)$, according to Eq.(28), we have

$$|\hat{x}_t \in p_t(x) \wedge \hat{x}_t \in \mathcal{B}(x_t, r_0)| \geq mc_t \pi_d r_0^d > \frac{2k_t}{n_t}. \tag{29}$$

Since $\hat{x}_t \in p_t(x)$ are independently drawn from the target distribution, let $\mathbb{I}(\cdot)$ to be the Indicator funciton and $S_{n_t}(x_t) = \sum_{i=1}^{n_t} \mathbb{I}(\hat{x}_t \in \mathcal{B}(x_t, r_0))$ denote the number of data $\hat{x}_t \in p_t(x)$ that fall into $\mathcal{B}(x_t, r_0)$, then $S_{n_t}(x_t)$ follows the Binomial distribution. Let $W \sim Binomial(n_t, \frac{2k}{n_t})$, according to the Chernoff inequality, we have

$$P(S_{n_t}(x_t) < k_t) \leq P(W < k_t)$$
$$= P(W - \mathbb{E}[W] < -k_t)$$
$$\leq \exp(-k_t^2/2\mathbb{E}[W])$$
$$= \exp(-k_t/4), \tag{30}$$

where the second inequality holds since $S_n(x)$ has a larger mean than $W$. With a large $k_t$, the probability that $S_n(x) < k_t$ is small for any $x_t \in p_t(x)$. Denoting $\hat{x}_t^{(i)}$ as the $i_{th}$ nearest sample to $x_t$ among $\mathcal{B}(x_t, r_0)$ in the cache, we have for any $x_t \in p_t(x)$

$$P(\| \hat{x}_t^{(k_t)} - x_t \| \le r_0) = P(S_n(x_t) \ge k_t) \ge 1 - \exp(-k_t/4) \tag{31}$$

Combine Eq.(31) with the assumption that the distribution $p_t(x)$ is finite with cardinality $\aleph_{p_t}$ and the desired probability part is shown by union bound.

$$\begin{aligned}
\bigcap_{x_t \in p_t(x)} P(\| \hat{x}_t^{(k_t)} - x_t \| \le r_0)) &= \bigcap_{x_t \in p_t(x)} P(S_n(x) \ge k_t) \\
&= 1 - \bigcup_{x_t \in p_t(x)} P(S_n(x) < k_t) \\
&\ge 1 - \aleph_{p_t} \exp\left(-\frac{k_t}{4}\right) \\
&= 1 - \exp\left(-\frac{k_t}{4} + \log \aleph_{p_t}\right).
\end{aligned} \tag{32}$$

And then we have the following proposition.

**Proposition 5.** *Given the target domain distributions $p_t(x)$ that is finite with cardinality $\aleph_{p_t}$, and $\Omega := \bigcup_{x \in p_t(x)} \mathcal{B}(x, r)$, where $\mathcal{B}(x, r) = \{x' : \| x' - x \| \le r\}$ denotes a ball centered on $x$ with radius $r$. Denote $f^* = \arg\min_{f \in \mathcal{H}}(\epsilon_t(f) + \epsilon_\Omega(f))$ and $\xi = \epsilon_t(f^*) + \epsilon_\Omega(f^*)$. Assume all hypotheses $h$ are $L$-Lipschitz continuous, the risk of hypothesis $\hat{f}$ on the unseen target domain is then bounded by*

$$\epsilon_t(\hat{f}) \le \kappa + \epsilon_\Omega(\hat{f}) + 2L\left(\frac{2k_t}{mc_t \pi_d n_t}\right)^{1/d}. \tag{33}$$

*with probability $1 - \exp(-\frac{k_t}{4} + \log \aleph_{p_t})$*

### C.2.3 Excess Error Bound of Cache Classifier

Let $s_i$ to be the softmax probability $softmax(\boldsymbol{p}_{cache})$ for class $i$ in the the cache classifier from Eq.(5), we can simplify the classifier as $\hat{f}_{cache} = \mathbb{I}\{s_1 \ge \frac{1}{2}\}$ on the binary classification setting. Then $\hat{f}_{cache}(x_t) \ne f^*(x_t)$ implies that $\left|\hat{f}_{cache}(x_t) - f^*(x_t)\right| \ge \left|f^*(x_t) - \frac{1}{2}\right|$. We then bridge the gap between the excess error and the classify error as follows:

$$\mathcal{E}_t(\hat{f}) = 2\mathbb{E}_{x_t \sim p_t(x)}\left[\left|f^*(x_t) - \frac{1}{2}\right| \mathbb{I}\left\{\left|\hat{f}_{cache}(x_t) - f^*(x_t)\right| \ge \left|f^*(x_t) - \frac{1}{2}\right|\right\}\right]. \tag{34}$$

We want to bound $\sup_{x_t} \left|\hat{f}_{cache}(x_t) - f^*(x_t)\right| \le t$, combining with the marginal assumption in Assumption 3 and the fact that

$$\mathbb{E}\left[Z \cdot \mathbb{I}\{Z \le t\}\right] \le t P(Z \le t), \tag{35}$$

where $Z = \left|f^*(x_t) - \frac{1}{2}\right|$, so we have $\mathcal{E}_t(\hat{f}) \le C_\beta t^{\beta+1}$. To bound $\left|\hat{f}_{cache}(x_t) - f^*(x_t)\right|$, we denote $(\hat{x}_t^{(i)}, \hat{y}_t^{(i)})$ as the $i_{th}$ nearest data and the corresponding labels to $x_t$ in $\mathcal{B}(x_t, r_0)$. The result of the cache classfier with normalized weight will be

$$\hat{f}_{cache}(x_t) = \sum_{i=1}^{k_t} \frac{1}{\sum_{j=1}^{k_t}\left[g\left(\hat{x}_t^{(j)}\right)^T g(x)\right]} \left[g\left(\hat{x}_t^{(i)}\right)^T g(x)\right] \hat{y}_t^{(i)} \tag{36}$$

$$= \sum_{i=1}^{k_t} w_i \hat{y}_t^{(i)}, \tag{37}$$

where $w_i = \frac{g\left(\hat{x}_t^{(i)}\right)^T g(x)}{\sum_{j=1}^{k_t}\left[g\left(\hat{x}_t^{(j)}\right)^T g(x)\right]}$ is the normalized weight and $\sum_{i=1}^{k_t} w_i = 1$. Based on the assumptions and notions above, we have for any $x_t \in p_t(x)$

$$\left| \hat{f}_{cache}(x_t) - f^*(x_t) \right| = \left| \sum_{i=1}^{k_t} w_i \hat{y}_t^{(i)} - f^*(x_t) \right|$$

$$\leq \left| \sum_{i=1}^{k_t} w_i \hat{y}_t^{(i)} - \sum_{i=1}^{k_t} w_i f^* \left( \hat{x}_t^{(i)} \right) \right| + \left| \sum_{i=1}^{k_t} w_i f^* \left( \hat{x}_t^{(i)} \right) - f^*(x_t) \right| \quad (38)$$

$$\leq \underbrace{\left| \sum_{i=1}^{k_t} w_i \hat{y}_t^{(i)} - \sum_{i=1}^{k_t} f^* \left( \hat{x}_t^{(i)} \right) \right|}_{①} + \underbrace{\sum_{i=1}^{k_t} w_i \left| f^* \left( \hat{x}_t^{(i)} \right) - f^*(x_t) \right|}_{②},$$

where ② is easy to bound. According to the assumption that $f^*$ is $C$-Smoothness, we have

$$\sum_{i=1}^{k_t} w_i \left| f^* \left( \hat{x}_t^{(i)} \right) - f^*(x_t) \right| \leq \sum_{i=1}^{k_t} w_i C \cdot \| \hat{x}_t^{(i)} - x_t \| \leq C \cdot \| \hat{x}_t^{(k_t)} - x_t \| \quad (39)$$

According to Eq.(31), with probability at least $1 - \exp(-k_t/4)$, ② $\leq C \left( \frac{2k_t}{mc_t \pi_d n_t} \right)^{1/d}$. Note that We store the target sample into the cache only when its prediction confidence is large enough. Therefore, it is natural to assume that:

$$E_{Y|X} \left[ \hat{y}_t^{(i)} \right] = f^*(x_t^{(i)}). \quad (40)$$

Then we use the Hoeffding inequality to obtain the upper bound of ①

$$P_{X,Y} \left( \left| \sum_{i=1}^{k_t} w_i \hat{y}_t^{(i)} - \sum_{i=1}^{k_t} f^*(\hat{x}_t^{(i)}) \right| > \epsilon \right)$$

$$= \mathbb{E}_X \left[ P_{Y|X} \left( \left| \sum_{i=1}^{k_t} w_i \hat{y}_t^{(i)} - \sum_{i=1}^{k_t} f^*(\hat{x}_t^{(i)}) \right| > \epsilon \right) \right]$$

$$\leq 2 \exp(-\frac{2\epsilon^2}{\sum_{i=1}^{k_t} w_i^2})$$

$$\approx 2 \exp(-2\eta k_t \epsilon^2). \quad (41)$$

We simplify the bound by assuming that the weights in the target domain are evenly distributed in the subset of all samples with respect to a specific class controlled by coefficient $\eta$, according to Assumption 1 and Proposition 4. That is, we have $\sum_{i=1}^{k_t} w_i^2 \approx \sum_{i=1}^{\eta k_t} \left( \frac{1}{\eta k_t} \right)^2 = \frac{1}{\eta k_t}$.

Set $\epsilon = (1/k_t)^{1/4}$, we have, with probability, at least $1 - 3 \exp(-2\eta \sqrt{k_t})$, ① $\leq (1/k_t)^{1/4}$, ② $\leq C \left( \frac{2k_t}{mc_t \pi_d n_t} \right)^{1/d}$, and then $\left| \hat{f}_{cache}(x_t) - f^*(x_t) \right| \leq (1/k_t)^{1/4} + C \left( \frac{2k_t}{mc_t \pi_d n_t} \right)^{1/d}$. According to Eq.(31) and Eq.(35), the excess error is bounded by

$$\mathcal{E}_t(\hat{f}) \leq 2C_\beta \left( \left( \frac{1}{k_t} \right)^{1/4} + C \left( \frac{2k_t}{mc_t \pi_d n_t} \right)^{1/d} \right)^{1+\beta}$$

$$\approx \left( \left( \frac{1}{k_t} \right)^{1/4} + C_1 \left( \frac{k_t}{c_t n_t} \right)^{1/d} \right)^{1+\beta}, \quad (42)$$

with constant $C_1$. When appropriately choosing $k_t = \mathcal{O}(\log n_t)$, we have

$$\min\{1 - 2\exp(-2\eta \sqrt{k_t}), 1 - \exp(-k_t/4)\}$$
$$\geq 1 - 2\exp(-2\eta \sqrt{k_t}) - \exp(-k_t/4)$$
$$\geq 1 - 3\exp(-2\eta \sqrt{k_t}) \quad (43)$$
$$= 1 - 3\exp(-\mathcal{O}(1)\sqrt{\log n_t})$$

where the third line is because $k_t/4 > 2\eta\sqrt{k_t}$ for large enough $k_t$. Namely, with probability at least $1 - 3\exp(-\sqrt{\log n_t})^{\mathcal{O}(1)}$, the following bound holds true.

$$\mathcal{E}_t(\hat{f}) \leq \mathcal{O}\left(\left(\frac{1}{\log n_t}\right)^{1/4} + \left(\frac{\log n_t}{c_t n_t}\right)^{1/d}\right)^{1+\beta}, \tag{44}$$

### C.3   Historical Cache benefits from Boosting Samples (Proof of Proposition 3)

To study the effect of the boosting samples, we consider the cache classfier containing both $k_t$ historical samples $\{\hat{x}_t^{(i)}, \hat{y}_t^{(i)}\}_{i=1}^{k_t}$ and $k_b$ boosting samples $\{\hat{x}_b^{(i)}, \hat{y}_b^{(i)}\}_{i=1}^{k_b}$ as the nearest data to $x_t$ in $\mathcal{B}(x_t, r_0)$. With the normalized weights $w_{ti} = \frac{g(\hat{x}_t^{(i)})^T g(x)}{\sum_{j=1}^{k_t}\left[g(\hat{x}_t^{(j)})^T g(x)\right] + \sum_{j=1}^{k_b}\left[g(\hat{x}_b^{(j)})^T g(x)\right]}$ and $w_{bi} = \frac{g(\hat{x}_b^{(i)})^T g(x)}{\sum_{j=1}^{k_t}\left[g(\hat{x}_t^{(j)})^T g(x)\right] + \sum_{j=1}^{k_b}\left[g(\hat{x}_b^{(j)})^T g(x)\right]}$, the prediction result of the cache classifier will be $\hat{f}_{cache}(x_t) = \sum_{i=1}^{k_t} w_{ti}\hat{y}_t^{(i)} + \sum_{i=1}^{k_b} w_{bi}y_b^{(i)}$. Then we have:

$$
\begin{aligned}
&\left|\hat{f}_{cache}(x_t) - f^*(x_t)\right| \\
&= \left|\sum_{i=1}^{k_t} w_{ti}\hat{y}_t^{(i)} - \sum_{i=1}^{k_t} w_{ti}f^*(x_t) + \sum_{i=1}^{k_b} w_{bi}y_u^{(i)} - \sum_{i=1}^{k_b} w_{bi}f^*(x_t)\right| \\
&\leq \left|\left[\sum_{i=1}^{k_t} w_{ti}\hat{y}_t^{(i)} - \sum_{i=1}^{k_t} w_{ti}f^*(\hat{x}_t^{(i)})\right] + \left[\sum_{i=1}^{k_t} w_{ti}f^*(\hat{x}_t^{(i)}) - \sum_{i=1}^{k_t} w_{ti}f^*(x_t)\right]\right. \\
&\quad \left. + \left[\sum_{i=1}^{k_b} w_{bi}y_u^{(i)} - \sum_{i=1}^{k_b} w_{bi}f^*\left(x_u^{(i)}\right)\right] + \left[\sum_{i=1}^{k_b} w_{bi}f^*\left(x_u^{(i)}\right) - \sum_{i=1}^{k_b} w_{bi}f^*(x_t)\right]\right| \\
&\leq \underbrace{\left|\sum_{i=1}^{k_t} w_{ti}\hat{y}_t^{(i)} + \sum_{i=1}^{k_b} w_{bi}y_u^{(i)} - \sum_{i=1}^{k_t} w_{ti}f^*(\hat{x}_t^{(i)}) - \sum_{i=1}^{k_b} w_{bi}f^*\left(x_u^{(i)}\right)\right|}_{①} \\
&\quad + \underbrace{\sum_{i=1}^{k_t} w_{ti}\left|f^*(\hat{x}_t^{(i)}) - f^*(x_t)\right|}_{②} + \underbrace{\sum_{i=1}^{k_b} w_{bi}\left|f^*\left(x_u^{(i)}\right) - f^*(x_t)\right|}_{③}
\end{aligned}
$$

Similar to Eq.(40), we have the following assumption on the boosting distribution:

$$E_{Y|X}\left[\hat{y}_b^{(i)}\right] = f^*(x_b^{(i)}). \tag{45}$$

According to Eq.(41), we have

$$
\begin{aligned}
&P_{X,Y}\left(\left|\sum_{i=1}^{k_t} w_{ti}\hat{y}_t^{(i)} + \sum_{i=1}^{k_b} w_{bi}y_b^{(i)} - \sum_{i=1}^{k_t} w_{ti}f^*(\hat{x}_t^{(i)}) - \sum_{i=1}^{k_b} w_{bi}f^*\left(x_b^{(i)}\right)\right|\right) \\
&= \mathbb{E}_X\left[P_{Y|X}\left(\left|\sum_{i=1}^{k_t} w_{ti}\hat{y}_t^{(i)} + \sum_{i=1}^{k_b} w_{bi}y_b^{(i)} - \sum_{i=1}^{k_t} w_{ti}f^*(\hat{x}_t^{(i)}) - \sum_{i=1}^{k_b} w_{bi}f^*\left(x_b^{(i)}\right)\right|\right)\right] \\
&\leq 2\exp(-2\eta(k_t + k_b)\epsilon^2)
\end{aligned} \tag{46}
$$

Set $\epsilon = (1/(k_t + k_b))^{1/4}$, we have, with probability, at least $1 - 3\exp(-2\eta\sqrt{(k_t + k_b)})$, ① $\leq (1/(k_t + k_b))^{1/4}$. Then, according to Eq.(39), we have

$$\sum_{i=1}^{k_t} w_{ti} \left| f^*\left(\hat{x}_t^{(i)}\right) - f^*(x_t) \right| \leq \sum_{i=1}^{k_t} w_{ti} C \cdot \| \hat{x}_t^{(i)} - x_t \| \leq S_t C \cdot \| \hat{x}_t^{(k_t)} - x_t \| \tag{47}$$

and

$$\sum_{i=1}^{k_b} w_{bi} \left| f^*\left(\hat{x}_b^{(i)}\right) - f^*(x_t) \right| \leq \sum_{i=1}^{k_b} w_{bi} C \cdot \| \hat{x}_b^{(i)} - x_t \| \leq S_b C \cdot \| \hat{x}_b^{(k_b)} - x_t \| . \tag{48}$$

where $S_t = \sum_{i=1}^{k_t} w_{ti}$, $S_b = \sum_{i=1}^{k_b} w_{bi}$ are the sum of weights of historical samples and boosting samples, respectively, and we have $S_t + S_b = 1$.

Then we have the following results in similar:

$$② \leq S_t C \left(\frac{2k_t}{mc_t\pi_d n_t}\right)^{1/d}; \quad ③ \leq S_b C \left(\frac{2k_b}{mc_b\pi_d n_b}\right)^{1/d} \tag{49}$$

Finally, the excess error under the covariate shift setting can be bounded by

$$\mathcal{E}_t(\hat{f}) \leq 2C_\beta \left( (\frac{1}{k_t + k_b})^{1/4} + S_t C \left(\frac{2k_t}{mc_t\pi_d n_t}\right)^{1/d} + S_b C \left(\frac{2k_b}{mc_b\pi_d n_b}\right)^{1/d} \right)^{1+\beta}$$

$$\approx \left( \left(\frac{1}{k_t + k_b}\right)^{1/4} + C_1 S_t \left(\frac{k_t}{c_t n_t}\right)^{1/d} + C_1 S_b \left(\frac{k_b}{c_b n_b}\right)^{1/d} \right)^{1+\beta} \tag{50}$$

$$= \left( \left(\frac{1}{k_t + k_b}\right)^{1/4} + C_1 \sum_{i=1}^{k_t} w_{ti} \left(\frac{k_t}{c_t n_t}\right)^{1/d} + C_1 \sum_{i=1}^{k_b} w_{bi} \left(\frac{k_b}{c_b n_b}\right)^{1/d} \right)^{1+\beta}$$

Compared Eq.(50) to Eq.(42) and $S_t + S_b = 1$, it is easy to verify that

$$(S_t + S_b)C \left(\frac{2(k_t + k_b)}{mc_t\pi_d n_t}\right)^{1/d} - S_t C \left(\frac{2k_t}{mc_t\pi_d n_t}\right)^{1/d} - S_b C \left(\frac{2k_b}{mc_b\pi_d n_b}\right)^{1/d}$$

$$\geq S_b C \left(\frac{2k_t}{mc_t\pi_d n_t}\right)^{1/d} - S_b C \left(\frac{2k_b}{mc_b\pi_d n_b}\right)^{1/d} \tag{51}$$

In general, the boosting distribution is more close to the test sample than the target distribution and we have $c_b > c_t$. Thus the difference in Eq.(51) is then larger than 0, namely incorporating boosting samples into the memory bank, the excess error can be further reduced.

## D    More Experiments

**Independent Cache for Boosting Samples.**  In BoostAdater, due to the cost of augmentation, the number of boosting samples is relatively smaller than the number of historical samples. Therefore, we use a joint cache for storing both historical and boosting samples to facilitate intra-sample and inter-sample interactions. Table 10 and Table 11 study the influence of using an independent cache for the boosting samples. As can be observed from the results, BoostAdapter suffers from slight performance degradation due to the independent cache.

Table 10: **Independent cache for boosting samples on the OOD benchmark.**

|  | Imagenet-V2 | Imagenet-Sketch | Imagenet-A | Imagenet-R | Average |
|---|---|---|---|---|---|
| Independent Cache | 65.37 | 50.62 | **64.56** | **80.96** | 65.38 |
| Joint Cache | **65.51** | **51.28** | 64.53 | **80.95** | **65.57** |

Table 11: **Independent cache for boosting sample on the Cross-Domain Benchmark.**

| | Caltech | Pets | Cars | Flowers | Food101 | Aircraft | SUN397 | DTD | EuroSAT | UCF101 | *Average* |
|---|---|---|---|---|---|---|---|---|---|---|---|
| Independent Cache | 94.69 | 88.88 | 69.19 | **71.94** | 86.99 | 26.76 | 67.64 | 44.21 | 61.20 | 69.63 | 68.11 |
| Joint Cache | **94.77** | **89.51** | **69.30** | 71.66 | **87.17** | **27.45** | **68.09** | **45.69** | **61.22** | **71.93** | **68.68** |

**Different Augmentation for Boosting Samples.** We make use of random crop followed by random horizontal flip as augmentations for generating boosting samples. Additionally, we further explore the influences of different augmentations applied to the randomly cropped images. The comparison methods include: (i) Random Brighness: Randomly set the brighness of image from 50% to 150%. (ii) Random Auto Contrast: Apply auto contrast over image with probability $p = 0.5$. (iii) Random Rotate: Randomly rotate the image from -45 degree to 45 degree. (iv) Random Vertical Flip: Apply vertical flip over image with probability $p = 0.5$. (v) Random Horizontal Flip (BoostAdapter): Apply horizontal flip over image with probability $p = 0.5$. The results are presented in Table 12 and Table 13. The results indicate that random horizontal flipping outperforms other augmentation methods, primarily because the images generated from horizontal flips are closer to the original distribution when training CLIP.

Table 12: **Comparison of different augmentations on the OOD benchmark .** Default settings are marked in gray .

| | Imagenet-V2 | Imagenet-Sketch | Imagenet-A | Imagenet-R | Average |
|---|---|---|---|---|---|
| Random Brightness | 65.10 | 51.24 | 62.10 | **81.03** | 64.87 |
| Random Auto Contrast | 65.50 | 50.79 | 64.33 | 80.57 | 65.30 |
| Random Rotate | 61.14 | 47.67 | 60.83 | 78.15 | 61.95 |
| Random Vertical Flip | 63.39 | 49.67 | 60.77 | 78.55 | 63.10 |
| Random Horizontal Flip | **65.51** | **51.28** | **64.53** | 80.95 | **65.57** |

Table 13: **Comparison of different augmentations on the Cross-Domain Benchmark.** Default settings are marked in gray .

| | Caltech | Pets | Cars | Flowers | Food101 | Aircraft | SUN397 | DTD | EuroSAT | UCF101 | *Average* |
|---|---|---|---|---|---|---|---|---|---|---|---|
| Random Brightness | 94.60 | **89.70** | 69.28 | 71.70 | 86.88 | 26.67 | **68.24** | 45.57 | 61.63 | 71.45 | 68.57 |
| Random Auto Contrast | 94.48 | 89.67 | 69.33 | 71.90 | **87.24** | 27.39 | 68.16 | 45.51 | 61.67 | 71.77 | **68.71** |
| Random Rotate | 94.52 | 89.59 | 67.74 | 71.30 | 85.91 | 24.27 | 67.56 | 45.45 | 60.72 | 70.66 | 67.77 |
| Random Vertical Flip | **94.89** | 89.53 | 68.75 | **72.19** | 86.78 | 24.99 | 67.72 | 45.27 | **61.56** | 70.82 | 68.25 |
| Random Horizontal Flip | 94.77 | 89.51 | **69.30** | 71.66 | 87.17 | **27.45** | 68.09 | **45.69** | 61.22 | **71.93** | **68.68** |

# E   Additional Ablation Results

**Historical Samples and Boosting Samples.** We provide more ablation results of the historical and boosting samples on the Cross-Dataset benchmark in Table 14. The observation is consistent with the results in Table 3, showing that CLIP gains improvements from both historical and boosting samples. Furthermore, when applied to various downstream tasks, the importance of regional bootstrapping becomes more significant, as indicated by the gap between BoostAdapter and the variant that uses boosting samples only.

**Number of Augmented Views for Boosting Samples.** The complete results on the number of augmented views are presented in Table 15 and Table 16. With more augmented views, BoostAdapter is able to better extract the fine-grained information from the original test sample, achieving improved performance.

Table 14: **Ablation study on historical samples and boosting sample on the Cross-Domain Benchmark.**

| | Caltech | Pets | Cars | Flowers | Food101 | Aircraft | SUN397 | DTD | EuroSAT | UCF101 | *Average* |
|---|---|---|---|---|---|---|---|---|---|---|---|
| CLIP | 93.35 | 88.25 | 65.48 | 67.44 | 83.65 | 23.67 | 62.59 | 44.27 | 42.01 | 65.13 | 63.58 |
| Historical Samples | 94.16 | 89.42 | 66.87 | **72.11** | 85.93 | 24.69 | 67.24 | 44.80 | **61.85** | 69.81 | 67.69 |
| Boosting Samples | 94.32 | 88.64 | 68.38 | 71.54 | 87.12 | 27.30 | 67.42 | 44.68 | 45.93 | 69.34 | 66.47 |
| BoostAdapter | **94.77** | **89.51** | **69.30** | 71.66 | **87.17** | **27.45** | **68.09** | **45.69** | 61.22 | **71.93** | **68.68** |

Table 15: **Results of different views on the OOD benchmark.** Default settings are marked in gray.

| | Imagenet-V2 | Imagenet-Sketch | Imagenet-A | Imagenet-R | Average |
|---|---|---|---|---|---|
| 16 Views | 79.41 | 49.01 | 62.08 | 63.68 | 63.54 |
| 32 Views | 80.32 | 50.73 | 63.22 | 64.91 | 64.80 |
| 64 Views | 80.95 | 51.28 | 64.53 | 65.51 | 65.57 |
| 128 Views | 80.95 | 51.91 | 64.06 | 65.27 | 65.55 |

Table 16: **Results of different views on the Cross-Domain Benchmark.** Default settings are marked in gray.

| | Caltech | Pets | Cars | Flowers | Food101 | Aircraft | SUN397 | DTD | EuroSAT | UCF101 | *Average* |
|---|---|---|---|---|---|---|---|---|---|---|---|
| 16 Views | 93.95 | 89.62 | 68.06 | 71.62 | 86.76 | 25.71 | 67.33 | 45.39 | 62.07 | 70.97 | 68.15 |
| 32 Views | 94.48 | 89.59 | 69.07 | 71.54 | 87.01 | 27.18 | 67.97 | 45.45 | 61.22 | 71.56 | 68.51 |
| 64 Views | 94.77 | 89.51 | 69.3 | 71.66 | 87.17 | 27.45 | 68.09 | 45.69 | 61.22 | 71.93 | 68.68 |
| 128 Views | 94.77 | 89.62 | 69.15 | 71.34 | 87.28 | 27.15 | 68.15 | 45.86 | 61.19 | 71.87 | 68.64 |

**Fixed shot capacity.** We search the optimal total shot capicity in BoostAdapter. We also find that fixing the cache size to be 3 can generalize well in different task settings, as shown in Table 17 and Table 18.

# F  More Qualitative Results

More qualitative results are provided in Fig. 6.

Table 17: **Results of fixed shot capacity on the OOD benchmark.**

|  | Imagenet-V2 | Imagenet-Sketch | Imagenet-A | Imagenet-R | Average |
|---|---|---|---|---|---|
| CLIP | 60.86 | 46.09 | 47.87 | 73.98 | 57.20 |
| CLIP+TPT | 64.35 | 47.94 | 54.77 | 77.06 | 60.81 |
| PromptAlign | 65.29 | 50.23 | 59.37 | 79.33 | 63.55 |
| TDA | 64.67 | 50.54 | 60.11 | 80.24 | 63.89 |
| BoostAdapter-Fixed | **65.13** | **50.66** | 63.96 | 80.44 | 65.05 |
| BoostAdapter-Search | **65.03** | **50.66** | **64.27** | **80.64** | **65.15** |

Table 18: **Results of fixed shot capacity on the Cross-Domain Benchmark.**

|  | Caltech | Pets | Cars | Flowers | Food101 | Aircraft | SUN397 | DTD | EuroSAT | UCF101 | *Average* |
|---|---|---|---|---|---|---|---|---|---|---|---|
| CLIP | 93.35 | 88.25 | 65.48 | 67.44 | 83.65 | 23.67 | 62.59 | 44.27 | 42.01 | 65.13 | 63.58 |
| CLIP+TPT | 94.16 | 87.79 | 66.87 | 68.98 | 84.67 | 24.78 | 65.50 | 47.75 | 42.44 | 68.04 | 65.10 |
| PromptAlign | 94.01 | **90.76** | 68.50 | **72.39** | 86.65 | 24.80 | 67.54 | 47.24 | 47.86 | 69.47 | 66.92 |
| TDA | 94.24 | 88.63 | 67.28 | 71.42 | 86.14 | 23.91 | 67.62 | **47.40** | 58.00 | 70.66 | 67.53 |
| BoostAdapter-Fixed | **94.77** | 88.85 | **69.30** | 71.66 | 87.17 | 27.00 | 67.64 | 44.33 | **61.22** | 69.73 | 68.17 |
| BoostAdapter-Search | **94.77** | 89.51 | **69.30** | 71.66 | **87.17** | 27.45 | 68.09 | 45.69 | 61.22 | **71.93** | **68.68** |

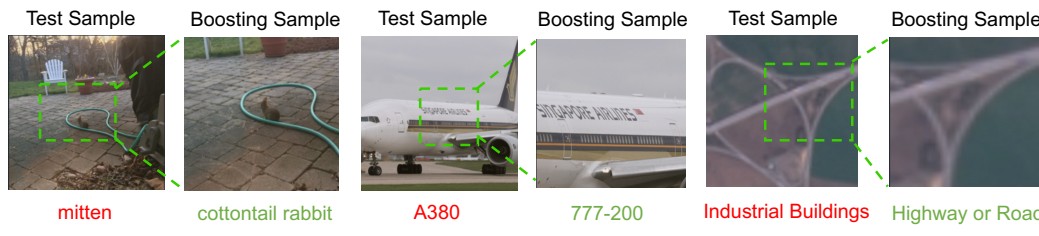

Figure 6: More qualitative results on ImagNet-A, Aircraft and EuroSAT.

