# OpenReview forum: "BoostAdapter: Improving Vision-Language Test-Time Adaptation via Regional Bootstrapping"
_NeurIPS.cc/2024/Conference — NeurIPS 2024 poster_

### Official Review · Reviewer_HZ6e · 2024-07-09

**Soundness:** 3
**Presentation:** 3
**Contribution:** 2
**Rating:** 6
**Confidence:** 3

**Summary:**

This paper proposes a training-free test-time adaptation approach for vision-language models. It combines the idea of entropy minimization with a training-free adaptor to enhance adaptation performance. The experimental results generally demonstrate the effectiveness of the proposed method.

**Strengths:**

1. The paper effectively combines entropy minimization for storing sample information and a training-free adaptor for test-time adaptation on VLMs. This approach proves to be effective in most cases.
2. The figures presented in the paper are clear and easy to follow. The performance figure, in particular, is very intuitive.
3. The method is theoretically supported, which adds credibility to the proposed approach.
4. The experiments are well-conducted and cover a broad range of considerations.

**Weaknesses:**

1. While the proposed method is effective, it lacks significant novel ideas or insights. The contributions could be seen as incremental rather than groundbreaking.
2. **Figure 2 (b):** The double-arrow directions in Figure 2 (b) should be colored in green and orange for clarity. This part is confusing.
3. **Figure 3:** The term "Boosting Cache" in Figure 3, which appears to store both boosting samples and historical samples, is not explained or used elsewhere in the manuscript, leading to confusion.
4. Several typographical errors need to be corrected.
5. The paper misses parameter study such as the threshold $\tau$.
6. In Figure 4b, the setup for how the model performs entropy minimization, updates learnable prompts or updates the LN/FULL model is missing. This information is crucial for understanding the ablation study.
7. The "Hand-crafted Prompt" is used in this paper, but it is unclear what it entails. Additionally, it is important to investigate if the method maintains stable performance across different prompts.
8. I am also wondering if the method could be categorized into TTA, as it maintains low-entropy samples per class in the cache. Even with such extra information for one or two, it cannot surpass the zero-shot clip. (Figure 4.c)

**Questions:**

1. What unique insights or techniques does your approach introduce that differentiate it from existing methods?
2. Why the proposed BoostAdaptor cannot perform as well as clip when the total shot capacity is less than 3? Is this a general phenomenon for all datasets?
3. Could you explain why the independent cache is good for some datasets as shown in Table 7?

**Limitations:**

1. While the method is effective, the paper lacks significant novel insights or groundbreaking ideas. The combination of entropy minimization and a training-free adaptor, though practical, may be viewed as an incremental improvement rather than a major innovation.
2. Refer to the weakness and question part.

---

> ### Author Rebuttal · Authors · 2024-08-06
>
> **Q1, Q9, Q12. Technical insights and difference with existing baselines.**
>
> Please refer to Q1 in Global Response.
>
> **Q2, Q3, Q4. Figure 2 (b), Figure 3 and typo.**
>
> Thanks for your advice. We will revise the figures, rewrite the corresponding part, and fix the typo in the revision.
>
> **Q5. Ablation study on threshold $\tau$.**
>
> We follow the setting in TPT [1] and empirically set the threshold $\tau=0.1$. To have a better understanding of the threshold, we provide the ablation results on the Aircraft dataset in Figure 1 of the rebuttal PDF.
>
> The results are consistent with the conclusion from Figure 4(b) in the original TPT paper, showing that a threshold near $0.1 \sim 0.2$ contributes to the best results, while higher thresholds lead to noisy samples that may result in misleading predictions.
>
> With the default setting that utilizes 64 augmented views, we can find that $6$ high-quality boosting samples are sufficient to bring significant improvements to the cache models.
>
>
> **Q6. More details about entropy minimization.**
>
> When implementing entropy minimization of the training-required methods, we follow the pipeline of TPT [1] and update the learnable prompts in the input while freezing other weights of the model. Taking entropy as a self-supervised objective, we perform gradient descent over both historical and boosting samples. Prompt tuning is more lightweight than full model tuning but still requires a large computational cost during model optimization.
>
> It is observed from Figure 4 (b) of the manuscript that either training-required or training-free methods benefit from both historical and boosting samples, in line with the theoretical analysis provided in Propositions 2 and 3. This further highlights our contribution to bridging the gap between training-required and training-free methods.
>
> Thanks for your advice and we will modify these parts to provide a clearer description.
>
>
> **Q7. Hand-crafted Prompt.**
>
> We follow the pipeline of TDA and adopt hand-crafted prompts in the training-free adapters. Take the Action Recognition dataset UCF101 for instance, TDA utilizes the prompt "a photo of a person doing \{\}." to better incorporate the prior knowledge of the dataset into the model.
>
> To study the influence of these hand-crafted prompts, we further equip CLIP, TDA, and BoostAdapter with different prompts (the standard prompt "a photo of \{\}." and the hand-crafted prompt "a photo of a person doing \{\}.") and compare their performance on the UCF101 dataset.
>
> As can be seen from the results in Figure 3 of the rebuttal PDF, BoostAdapter surpasses TDA and CLIP with both the standard prompt and the hand-crafted prompt. Additionally, the hand-crafted prompt brings improvements in performance to all the methods to some extent.
>
>
> **Q8, Q10. Comparison with zero-shot clip when the shot capacity is less than 3.**
>
> Our methods could be categorized into TTA since it adaptively makes predictions for streaming data during test time based on feature retrieval of historical and boosting samples.
>
> Besides, the reviewer points out that BoostAdapter cannot perform as well as CLIP when the shot capacity is less than 3. However, this is not always the case. As shown in Figure 2 of the rebuttal PDF, BoostAdapter outperforms CLIP across all 4 tasks on the OOD benchmarks. On the Aircraft dataset, the low-entropy samples stored in the boosting cache with a small shot capacity may be biased and not diverse enough, thereby leading to the performance drop.
>
>
> **Q11. Independent cache v.s. Joint cache.**
>
> The independent cache will retain all the knowledge from both historical and boosting samples, whereas in the joint cache, we update the cache of historical samples with boosting samples. Due to the limited cache size, the historical samples in the joint cache will be replaced by lower-entropy boosting samples when necessary.
>
> In most cases, BoostAdapter performs better with the joint cache compared to the independent cache. However, in some cases, when the test sample benefits from a sufficient amount of diverse cross-sample interactions, the replacement of boosting samples in the joint cache may lead to a slight performance drop.
>
> Generally, it is preferred to utilize the joint cache rather than the independent cache due to lower storage cost and better performance on average.
>
> **Reference**
>
> [1] Shu, Manli, et al. "Test-time prompt tuning for zero-shot generalization in vision-language models." Advances in Neural Information Processing Systems 35 (2022): 14274-14289.

---

> > ### Comment · Reviewer_HZ6e · 2024-08-09
> >
> > Thank you for your reply. I appreciate your additional experiments and explanation.
> >
> > Actually, after check out your rebuttal, I still feel confused about the Q1 and answer in your global reply. I've checked TDA and the idea of this paper is quite aligned with them. While discussing entropy minimization in TTA (and as you claimed ``Our main contribution lies in the theoretical and experimental connection between training-required and training-free methods.''), for me, it seems like combining effective training-free and training-required techniques.
> >
> > In this case, I would like to explain this paper in the way:
> > 1. Using a Tip-adapter-like (also TDA-like) training-free adapter.
> > 2. Using a memory bank to save reliable (historical and ) and diverse (augmented) test samples, using clip prediction as evidence.
> > 3. Apply this stored information to ``correct'' the clip logits for final prediction.
> > Please correct me if I have any misunderstanding.
> >
> > I am also confused about the relationship between historical and boosting samples. Will the boosting samples constructed by augmented filtered historical test samples? If this is the case, I saw so many similar ideas in the online test-time adaptation on pages 12-13 [R1]. Could you explain further about the insights of your strategy and why it is better than others? How to differentiate your contribution among them?
> >
> > It will be better to make the whole sample saving or caching process a bit clearer. In the current version, it is hidden among the equations.
> >
> > Another question is, could you specify why using EATA as the TTA baseline? EATA has carefully designed the FIM module and used the source sample information. It is also better for continual TTA. Why don't use other TTA methods?
> >
> > Thank you.
> >
> > Reference:
> >
> > [R1]. Wang, Z., Luo, Y., Zheng, L., Chen, Z., Wang, S., & Huang, Z. (2023). In search of lost online test-time adaptation: A survey. arXiv preprint arXiv:2310.20199.

---

> > > ### Author Response · Authors · 2024-08-10
> > >
> > > We thank the reviewer for the insightful comments, and we are happy to discuss some implementation details.
> > >
> > > **Q1. The implementation steps of BoostAdapter.**
> > >
> > > Your description of the implementation steps is correct. We would like to add some noteworthy points.
> > >
> > > - The boosting cache (memory bank) in step 2 is instance-adaptive. We create a copy of the historical cache (referred to as the boosting cache) and construct boosting samples for current test image to update the this cache for feature retrieval.
> > > - These boosting samples will be discarded and will not be used by other images after step 3.
> > >
> > > It is rational because the boosting samples are close to the current test image rather than others, ensuring the bound of empirical risks in Proposition 3.
> > >
> > >
> > >
> > > **Q2. The relationship between historical and boosting samples**
> > >
> > > For the current test image, the boosting samples will be derived only from itself rather than from historical samples.
> > >
> > > We check the survey and find that all the relevent methods perform techniques like augmentation and clustering over **only historical samples** in the memory bank, without considering any information of the current test image.
> > >
> > > We would like to point out that these methods may show poor generalization performance especially in downstream tasks that require fine-grained knowledge or when historical samples share insufficient similarity.
> > >
> > > So we construct the instance-aware boosting cache to perform information mining over the current test sample and incorporate this knowledge with historical samples during feature retrieval.
> > >
> > > The survey provides a clear description of memory-bank-based methods, so we will mention it and rewrite the corresponding section for a detailed discussion with these methods in the revision.
> > >
> > >
> > >
> > > **Q3. Using EATA as additional training-required methods.**
> > >
> > > The online test time adaptation setting discussed in BoostAdapter can be seen as a special case of continual TTA since we deal with samples from the test data stream. Therefore, we previously use EATA [1] simply due to its importance in test time adaptation and its applicability to our tasks. In practice, we follow the idea of a diversity-based selective strategy proposed by EATA to construct boosting samples in the cache.
> > >
> > > We further incorporate techniques from more training-required methods, including the Pseudo-Label Probability Difference metric (PLPD) from DEYO [2] and the consistency filter from TSD [3]. Specifically, in the BoostAdapter+DEYO variant, we filter out augmented views with a PLPD lower than 0.2. For the BoostAdapter TSD variant, we discard augmented views that have different cache predictions and CLIP predictions to ensure consistency of the boosting samples. The results are provided in Table A and we can observe performance improvements with the help of different training-required methods, demonstrating the versatility of BoostAdapter.
> > >
> > >
> > >
> > > **TableA: Unification of more training-required methods.**
> > >
> > > |                       | -V    | -S    | -A    | -R    | Average |
> > > | --------------------- | ----- | ----- | ----- | ----- | ------- |
> > > | CLIP-ViT-B/16         | 60.86 | 46.09 | 47.87 | 73.98 | 57.20   |
> > > | TDA                   | 64.67 | 50.54 | 60.11 | 80.24 | 63.89   |
> > > | BoostAdapter          | 65.03 | 50.66 | 64.27 | 80.64 | 65.15   |
> > > | BoostAdapter+EATA [1] | 65.27 | 50.82 | 64.83 | 81.15 | 65.52   |
> > > | BoostAdapter+DEYO [2] | 65.51 | 51.01 | 64.57 | 81.11 | 65.55   |
> > > | BoostAdapter+TSD [3]  | 65.49 | 51.50 | 64.37 | 81.15 | 65.63   |
> > >
> > >
> > >
> > > **Reference**
> > >
> > > [1] Niu, Shuaicheng, et al. "Efficient test-time model adaptation without forgetting." International conference on machine learning. PMLR, 2022.
> > >
> > > [2] Lee, Jonghyun, et al. "Entropy is not enough for test-time adaptation: From the perspective of disentangled factors." The International Conference on Learning Representations. ICLR (2024).
> > >
> > > [3] Wang, Shuai, et al. "Feature alignment and uniformity for test time adaptation." Proceedings of the IEEE/CVF Conference on Computer Vision and Pattern Recognition. 2023.

---

> > > > ### Comment · Reviewer_HZ6e · 2024-08-11
> > > >
> > > > Thank you for your reply and your additional experiments. Although I feel like the novelty of this work is still kind of limited, I would like to increase my score due to your prompt and detialed reply.

---

> > > > > ### Author Response · Authors · 2024-08-12
> > > > >
> > > > > Thanks a lot for your efforts and discussion!

---

### Official Review · Reviewer_a236 · 2024-07-13

**Soundness:** 3
**Presentation:** 3
**Contribution:** 2
**Rating:** 4
**Confidence:** 4

**Summary:**

This paper studies the problem of test-time vision-language model adaptation. The authors devise training-free method by maintaining a key-value memory for feature retrieval from both historical and boosting samples. The boosting samples are drawn from regional bootstrapping and capture the knowledge of the test sample it self. Experiments demonstrate the effectiveness of the proposed method.

**Strengths:**

The studied training-free adaptation setting is practical, broadening the application scope of test-time adaptation in real-world scenarios.

The authors propose incorporating augmentations of each sample into the cache to enable the cache-based classifier to perform better on more fine-grained classifications. This approach is both interesting and technologically sound.

The authors also provide theoretical analyses to establish connections between training-required and training-free TTA methods.

**Weaknesses:**

It would be beneficial for the authors to discuss the detailed technical differences from TDA more thoroughly. From my perspective, the key difference appears to be the introduction of additional augmented views of the same sample into the Boosting Cache for intra-sample iteractions. If this is the case, the technical contribution of this work might be a bit limited.

The proposed method relies on multiple augmentations, requiring multiple forward passes to achieve better performance than TDA, which sacrifices efficiency. For Figure 4(a), could the authors provide results on more datasets (both OOD domains and cross domains) to demonstrate the sensitivity of the proposed method to this hyper-parameter? This would help verify whether the proposed method can achieve good performance with fewer augmentations across multiple datasets.

The improvement on Cross-Domain Benchmarks with RN-50 backbone is a bit marginal.

**Questions:**

Could the authors provide some computational complexity analyses and comparisons for the proposed method, including wall-clock time and GPU memory consumption?

It would also be much better to conduct the analysis in Figure 4(c) on more datasets.

Is the proposed method applicable to, or have the authors tested it on, pure CNN/ViT models? If not, I recommend including ‘vision-language model adaptation’ in the title.

The authors claim that *“prior methods like TDA only consider inter-sample interactions and may fail to generalize well when the downstream tasks require fine-grained knowledge or there is 109 insufficient similarity across samples.”* Although I acknowledge the technological soundness of the proposed method can perform better on more fine-grained classification, are there any empirical evidences to further justify this?

In the Boosting Cache, do the authors store the original samples or their corresponding features? Storing images may pose privacy and additional computation issues. It would be helpful to indicate this in Figure 3.

[minor] How about the performance of the proposed on corruption datasets such as ImageNet-C?

**Limitations:**

the potential limitation is the computational efficiency of the proposed method, which can be alleviated if the method works well on multiple datasets with a small number of augmentations

---

> ### Author Rebuttal · Authors · 2024-08-06
>
> **Q1. Technical insights and differences with existing baselines.**
>
> Please refer to Q1 in Global Response.
>
> **Q2, Q5. Computation overhead and efficiency.**
>
> Please refer to Q2 in Global Response.
>
> **Q3, Q11. Number of augmented views.**
>
> The analysis of the augmented views can be found in Table 12 of the Technical Appendix. It can be observed that BoostAdapter shows superior performance on average compared to TDA with 32 views on the OOD benchmark and only 16 views on the Cross-Domain Benchmark.
>
> We would like to point out that TDA also utilizes augmentation to obtain high-quality embeddings of test samples to store in the cache, as depicted in Table 1 of the rebuttal PDF. Furthermore, most of the existing TTA methods use 64 views of augmentation by default. Therefore, using 64 views is not considered particularly large and is acceptable from a computational cost perspective.
>
> **Q4. Cross-Domain Benchmarks results with RN-50 backbone.**
>
> The marginal improvement is mainly due to the limited performance of the RN-50 backbone.
> To verify the robustness of BoostAdapter, we further provide results with more checkpoints in Tables 2 and 3 of the rebuttal PDF. BoostAdapter shows promising improvements and consistently outperforms TDA in 7, 8, and 7 out of 10 tasks on the Cross-Domain Benchmarks with RN101, ViT-B/32, and ViT-L/14 backbones, respectively.
>
> **Q6. Shot capacity.**
>
> We provide ablation results of the shot capacity over all four datasets on the OOD benchmarks in Figure 4 of the rebuttal PDF. The results are consistent with the conclusion in the manuscript that the boosting cache will achieve a balance of diversity and relevance as the shot capacity increases.
>
> **Q7. Applicability on Vision-Only Backbones.**
>
> Yes. We believe it is possible to apply BoostAdapter to vision-only TTA, but transferring might be non-trivial since vision-language TTA and vision-only TTA are generally two distinct research sub-fields with different settings. Our paper focuses on the adaptation of the vision-language model, while addressing vision-only models could be considered for future work.
>
> We appreciate your suggestion on refining the title of this paper and will incorporate it in the revision.
>
> **Q8. Fine-grained knowledge.**
>
> We provide qualitative results in Figure 5 of the appendix and Figure 4 of the rebuttal PDF to investigate how BoostAdapter leverages boosting samples to extract fine-grained knowledge.
> As depicted in the figures, boosting samples with low entropy incorporate the prior of label systems to filter out the noisy parts of the test images and guide the model on where to focus.
> Most importantly, we only need to perform random cropping and random horizontal flipping to achieve this, making BoostAdapter more applicable in real-world scenarios.
>
> **Q9. Storage in Boosting Cache.**
>
> In practice, we store the features of both the historical and boosting samples to construct the key-value cache, which is privacy-preserving and time-efficient. Thanks for your advice, and we will modify the figure for better clarity in the revision.
>
> **Q10. ImageNet-C.**
>
> Please refer to Q3 in Global Response.

---

> ### Author Response · Authors · 2024-08-12
>
> Dear reviewer,
>
> We would like to thank you for your insightful feedback. We hope that your concerns are addressed with our rebuttal. As we are getting really close to the deadline of the discussion phase, please let us know if there are any further questions that need clarification.
>
> Many thanks,
>
> Authors

---

> ### Comment · Reviewer_a236 · 2024-08-12
> **Follow up from reviewer**
>
> Thanks for the authors’ response. Regarding the computational efficiency, could the authors provide more details of the experimental setup, including but not limited to the GPU, and batch size?
>
> For FPS, I am confused about why BoostAdapter (64 forward passes of image encoder) achieves 11.23 fps and CLIP only achieves 82.3 fps which only needs 1 forward propagation. Meanwhile, the memory consumption of BoostAdapter and CLIP is the same. Do you test Inference Speed (fps) using batch size 64 (64 views also equal to batch size 64), and test Memory using a different batch size?
>
> Moreover, the performance could also be directly reported in Table 1 of the PDF.

---

> > ### Author Response · Authors · 2024-08-12
> >
> > **Q1. More details**
> >
> > We follow the setting in TDA and deal with the test samples from the data stream one by one. Therefore, we cannot increase the batch size to handle multiple different test samples simultaneously, but only for different views of the same test sample. Thus, we augment 64 views of the test image as TDA does and use 64 as the batch size to obtain the corresponding features.
> >
> > In BoostAdapter, we utilize a simpler augmentation (random crop and random horizontal flip) than AugMix in TDA to save time. Additionally, we set num_workers=8 in the dataloader to leverage multiprocessing for acceleration. Furthermore, we perform feature retrieval over the stored features instead of images. The additional retrieval time compared to TDA comes from the operation of updating the cache with boosted samples.
> >
> > All our experiments are conducted with a 64-core Nvidia 3090 24GB GPU.
> >
> >
> >
> > **Q2. FPS and memory.**
> >
> > For a better view of the time consumption, we provide the average wall-clock time consumption of each component for 1000 samples in TableA. The total time can be mainly divided into three parts: data augmentation, model forwarding, and feature retrieval.
> >
> > - Regarding augmentation time, note that we set num_workers = 8 for the dataloader, so the augmentation takes up a small percentage of the total time. TDA and BoostAdapter take a similar amount of time since they both utilize 64 views of augmentations.
> > - The model forwarding of BoostAdapter utilizes approximately 8 times more time than CLIP. This is reasonable since we use parallel forward propagation for the 64 views instead of sequential forward propagation, so the difference in comsumption overhead will not be as large as 64 times. The only difference is the batch size of the model input (1 for CLIP and 64 for BoostAdapter).
> >
> > - We build a new cache from the historical cache and update it with boosting samples so the feature retrieval time of BoostAdapter takes slightly longer than TDA. Overall, the time consumption in feature retrieval is much smaller than model forwarding.
> >
> > **TableA. Computation cost of each component.**
> >
> > |              | Data Augmentation      | Model Forwarding        | Feature Retrieval      | Total                  |
> > | ------------ | ---------------------- | ----------------------- | ---------------------- | ---------------------- |
> > | CLIP         | -                      | 0.01208 seconds (100%)  | -                      | 0.01208 seconds (100%) |
> > | TDA          | 0.00116 seconds (1.4%) | 0.08065 seconds (96.6%) | 0.00167 seconds (2.0%) | 0.08348 seconds (100%) |
> > | BoostAdapter | 0.00098 seconds (1.1%) | 0.07989 seconds (91.5%) | 0.00644 seconds (7.4%) | 0.08731 seconds (100%) |
> >
> > The memory consumption of BoostAdapter is similar to TDA because both utilize 64 views of augmentation, and the model takes 64 views of the image (batch size 64) as input. This model forwarding part accounts for most of the memory usage. During feature retrieval, we store the features in the cache without significant memory overhead, even with the integration of boosting samples' features.
> >
> > In the online test-time adaptation setting, we deal with test samples from the data stream one by one, so we cannot increase the batch size by parallel modeling of different test samples. We only perform parallel forward propagation for different views of the same test sample. Therefore, changing the batch size here is not appropriate, as it corresponds to the number of augmented views for the current test sample.
> >
> >
> >
> > **Q3. Efficiency analysis table.**
> >
> > Thanks for your insightful advice, and we add the performance into the efficiency analysis in the TableB.
> >
> >   **TableB. Efficiency analysis with performance results.**
> >
> > |              | Augmentation                   | Views | Inference Speed (fps) | Memory (GB) | OOD Benchmarks Results | Cross-Domain Benchmarks Results |
> > | ------------ | ------------------------------ | ----- | --------------------- | ----------- | ---------------------- | ------------------------------- |
> > | CLIP         | -                              | -     | 82.3                  | 1.2         | 57.20                  | 63.58                           |
> > | TPT          | Augmix                         | 64    | 0.29                  | 4.5         | 60.81                  | 65.10                           |
> > | DiffTPT      | Diffusion                      | 64    | 0.10                  | 14.4        | 60.52                  | 66.92                           |
> > | TDA          | Augmix                         | 64    | 11.89                 | 1.2         | 63.89                  | 67.53                           |
> > | BoostAdapter | Rand. Crop & Rand. Horiz. Flip | 64    | 11.23                 | 1.2         | 65.15                  | 68.52                           |

---

> > > ### Comment · Reviewer_a236 · 2024-08-12
> > >
> > > Thanks for the clarifications. It gets clearer now.
> > >
> > > > "We only perform parallel forward propagation for different views of the same test sample."
> > >
> > > In this implementation, a further question concerning memory usage is: why do BoostAdapter and CLIP have the same reported memory consumption? Shouldn’t the parallel forward passes of 64 views in BoostAdapter require more memory than a single view forward propagation in CLIP?

---

> > > > ### Author Response · Authors · 2024-08-12
> > > >
> > > > Sorry for the confusion about the memory usage of CLIP.  It is reported by mistake when we compare the memory consumption of BoostAdapter with TDA. Actually, the memory consumption of CLIP is just 0.7 GB, which is lower than the 1.2 GB of BoostAdapter and TDA. We hope that your questions are addressed.

---

### Official Review · Reviewer_YJpm · 2024-07-16

**Soundness:** 3
**Presentation:** 3
**Contribution:** 2
**Rating:** 5
**Confidence:** 4

**Summary:**

The paper focuses on gradient-free test time adaptation of CLIP model with ViT-B/16 and ResNet-50 backbones on out-of-distribution datasets. The authors take inspiration of augmentations from gradient-based test time methods and incorporate this concept of augmentations in gradient-free and memory (cache) based test time methods. Previous work considers only historic samples to be in cache, whereas this work also includes the augmentations of test samples with lower entropy into the cache. Results on OOD benchmark and cross-domain benchmark show improved average performance compared to prior works.

**Strengths:**

1.	Proposed a simple but effective approach to include low entropy augmentations into the memory.

2.	Established theoretical bounds to justify the inclusion of augmentations into memory and its relation to minimization of empirical risk.

3.	Proposed approach brings significant improvements on ImageNet-A, Aircraft and EuroSAT datasets.

**Weaknesses:**

I don’t have major concerns on the proposed approach, as the method is simple and straightforward. My concern lies in increase of computation overhead and extended run time due to running CLIP model on multiple augmentations during test time (as acknowledged by the authors). In addition, it can be noticed that results are comparable on almost all datasets except ImageNet-A, Aircraft and EuroSAT datasets. It would be interesting to provide rationale for the proposed approach to work much better on these datasets. Results are shown on single ViT architecture, however providing results on multiple CLIP based ViT backbones would be interesting as the method targets for test time performance. It would be helpful to evaluate the method on more OOD distributions like corruptions (ImageNet-C) to understand it better.

**Questions:**

Please refer to Weakness above.

**Limitations:**

Limitations are discussed in the work.

---

> ### Author Rebuttal · Authors · 2024-08-06
>
> **Q1. Computation overhead and efficiency.**
>
> Please refer to Q2 in Global Response.
>
> **Q2. Specific Datasets.**
>
> As shown in Figure 5 in the appendix, BoostAdapter benefits from boosting samples to capture fine-grained knowledge of the test samples. We further provide more qualitative results on the ImageNet-A, Aircraft, and EuroSAT datasets in Figure 4 of the rebuttal PDF to investigate how BoostAdapter performs on these datasets.
>
> ImageNet-A consists of real-world examples that are misclassified by ResNet models, while Aircraft is a benchmark dataset for the fine-grained visual categorization of aircraft, and EuroSAT consists of satellite images for land use and land cover classification.
>
> These datasets require fine-grained information for classification, and the boosting samples filter out noisy parts of the test samples while retaining useful information, contributing to significant performance enhancements.
>
> **Q3. More backbones.**
>
> In order to further validate the robustness of our method, we compare BoostAdapter with baseline models across various backbones including both RN and ViT checkpoints.
>
> The results in Table 2 and Table 3 of the rebuttal PDF indicate that BoostAdapter shows strong compatibility and consistently outperforms TDA in most of the cases over different backbones. For instance, BoostAdapter shows superior performance to TDA over 7, 8, and 8 out of 10 tasks with RN-101, ViT-B/32, and ViT-L/14 checkpoints on the Cross-Domain Benchmark, respectively.
>
> **Q4. ImageNet-C.**
>
> Please refer to Q3 in Global Response.

---

> ### Author Response · Authors · 2024-08-12
>
> Dear reviewer,
>
> We would like to thank you for your insightful feedback. We hope that your concerns are addressed with our rebuttal. As we are getting really close to the deadline of the discussion phase, please let us know if there are any further questions that need clarification.
>
> Many thanks,
>
> Authors

---

> ### Comment · Reviewer_YJpm · 2024-08-13
>
> Dear authors,
>
> I thank you for the responses, and additional experiments provided in the rebuttal. I have read fellow reviewers comments and authors responses. I find that the method is slightly better than TDA consistently across all benchmarks, particularly helpful for ImageNet-A. However, my major concern on limited novelty still exists and hence I tend to keep a borderline rating. I will discuss with my fellow reviewers for a final decision.

---

### Author Rebuttal · Authors · 2024-08-06

We thank all reviewers for their valuable feedback and are encouraged by the positive comments on our contributions, including

1. Soundness and Novelty:
    - Training-free adaptation broadens real-world applicability (Reviewer a236).
    - Innovative use of sample augmentations in the cache for fine-grained classifications (Reviewer a236 & YJpm).

2. Theoretical Contribution:
    - Establishes connections and bounds between training-required and training-free TTA methods (Reviewer a236 & YJpm).
    - Adds credibility and justification to the approach (Reviewer HZ6e).

3. Solid Experiments:
    - Significant improvements on ImageNet-A, Aircraft, and EuroSAT datasets (Reviewer YJpm).
    - Well-conducted experiments cover a broad range of considerations (Reviewer HZ6e).

4. Presentation:
    - Clear and intuitive figures, especially the performance figure (Reviewer HZ6e).

In the following parts, we will first respond to the common questions raised by reviewers and then respond to the rest of the concerns of each reviewer from point to point. We believe the comments & revisions have made the paper stronger and thank all the reviewers for their help. ***Please let us know if these address your concerns and if there are any further questions that need clarification.***

**Q1. Technical insights and difference with TDA.**

The mainstream of training-required TTA methods is entropy minimization, while TDA serves as a training-free baseline that maintains a key-value cache of historical samples. We argue that our work is far more than merely introducing augmentation views into training-free adapters. **Our main contribution lies in the theoretical and experimental connection between training-required and training-free methods.**

Specifically, we provide a bound for empirical risk as a theoretical guarantee in unifying the two streams. We also show that both methods can benefit from each other, enhancing performance through either entropy minimization or feature retrieval of both historical and boosting samples, as illustrated in Figure 4(b) of the manuscript.
In practice, we focus on boosting training-free methods rather training-required ones due to their lower computational cost and better generalization performance.

**Additional evidence on this point:** From the unified perspective, we can also enhance training-free adapters with additional training-required methods. Here we take EATA [2], which introduces momentum statistics to combat forgetting in the test-time adaptation, as the showcase.
When equipping BoostAdapter with the technique of EATA, we observe further improvement and find that training-free adapters can benefit from various boosting techniques of training-required methods. These results can be found in Table 2 of our rebuttal PDF.


**Q2. Computation overhead and efficiency.**

We agree that efficiency is an important metric, and we have already provided the computational efficiency analysis in Table 16 in the Technical Appendix. More information is available in Table 1 of our rebuttal PDF.

We would like to point out that TDA follows TPT [1] and utilizes 64 views with AugMix to obtain high-quality embeddings of test samples, which leads to a similar computation cost with the boosting augmentation in BoostAdapter.
As can be seen from the results, the inference time for BoostAdapter is slightly longer than TDA but remains significantly faster than training-required methods such as TPT and DiffTPT.
Therefore, considering the performance enhancement it provides, the additional inference cost and comparable memory consumption of BoostAdapter are acceptable.


**Q3. More results on ImageNet-C.**

To further evaluate the generalization ability of BoostAdapter in new test-time scenarios, we compare BoostAdapter with baseline methods on the Imagenet-C dataset at the highest severity level 5. These results are presented in Table 5 of the rebuttal PDF.

The key observation from these results is that BoostAdapter consistently outperforms TDA across all 15 corruption types, highlighting its practical applicability in real-world situations.

BoostAdapter's superior performance stems from its capability to capture the knowledge of the test sample even under severe corruption. This is achieved with the help of the boosting samples, which effectively filter out noisy parts while retaining useful information in the images.

**Reference**

[1] Shu, Manli, et al. "Test-time prompt tuning for zero-shot generalization in vision-language models." Advances in Neural Information Processing Systems 35 (2022): 14274-14289.

[2] Niu, Shuaicheng, et al. "Efficient test-time model adaptation without forgetting." International conference on machine learning. PMLR, 2022.

---

### Comment · Area_Chair_33p5 · 2024-08-10
**Please engage in discussion**

Dear Reviewers,

The authors have provided responses to the reviews. Please take a look at the responses (and other reviews) and engage in a discussion.

Thanks for your service to NeurIPS 2024.

Best,
AC

---

### Decision · Program_Chairs · 2024-09-25

**Decision:**

Accept (poster)

**Comment:**

This paper introduces a training-free test-time adaptation method for vision-language models that proposes to include augmented samples that are predicted with low entropy into the memory bank. Theoretical justification is provided to show that these new samples reduce empirical risk. Reviewers thought the method was simple but well-justified theoretically, and that there were comprehensive experiments showing improvements. However, reviewers also had concerns about novelty, marginal performance improvements and increased computational overhead due to the use of multiple augmentations. There was a good discussion between reviewers and authors that resolved concerns regarding computational overhead though concerns about novelty remained.

The AC's view is that while in practice the differences with the closely related TDA method appear small, the conceptual contribution of linking gradient-based TTA and memory-based TTA methods, along with theoretical justification for the apparently simple addition are useful technical contributions. Thus, on balance, even though the improvements are somewhat small, the AC recommends acceptance.